# Time-compressed preplay of anticipated events in human primary visual cortex

Matthias Ekman[1], Peter Kok[1,2] & Floris P. de Lange[1]

Perception is guided by the anticipation of future events. It has been hypothesized that this process may be implemented by pattern completion in early visual cortex, in which a stimulus sequence is recreated after only a subset of the visual input is provided. Here we test this hypothesis using ultra-fast functional magnetic resonance imaging to measure BOLD activity at precisely defined receptive field locations in visual cortex (V1) of human volunteers. We find that after familiarizing subjects with a spatial sequence, flashing only the starting point of the sequence triggers an activity wave in V1 that resembles the full stimulus sequence. This preplay activity is temporally compressed compared to the actual stimulus sequence and remains present even when attention is diverted from the stimulus sequence. Preplay might therefore constitute an automatic prediction mechanism for temporal sequences in V1.

[1] Donders Institute For Brain, Cognition and Behaviour, Radboud University Nijmegen, Nijmegen 6500 HB, Netherlands. [2] Princeton Neuroscience Institute, Princeton University, Princeton, New Jersey 08540, USA. Correspondence and requests for materials should be addressed to M.E. (email: m.ekman@donders.ru.nl).

The visual system is predictive in nature, anticipating relevant events to facilitate sensory processing and decision-making[1]. Prediction in perception has often been studied in static contexts, where a stimulus is expected because the base rate of occurrence is higher[2] or because of statistical associations between stimuli[3]. These forms of prediction can be neurally implemented by pre-activating sensory representations of the expected events[4–6].

However, real-world predictions are typically dynamic: for example, we predict the trajectory of a ball moving towards us or whether a car will hit us if we cross the road. Implementing this kind of dynamic prediction is more complex, as it requires an anticipatory wave of visual responses that is both spatially and temporally precise. Recently, such waves of preplay activity have been observed in the visual cortical system of rats[7] and monkeys[8], but the existence, function and potential source of preplay waves in humans however remain unknown.

Here we tested whether human primary visual cortex (V1) is engaged in dynamic prediction by preplaying anticipated visual events. We characterized neural activity in the primary visual cortex at both high spatial and temporal resolution, by combining ultra-fast functional magnetic resonance imaging (fMRI, using a volume acquisition time (TR) of 88 ms) and population receptive field (pRF) mapping[9] to identify retinotopically specific responses with high temporal resolution.

We found that flashing only the starting point of a moving dot sequence triggered an activity wave in V1 that recreates the full stimulus sequence. This anticipatory activity wave was temporally compressed compared to the actual stimulus sequence and was present even when attention was diverted from the stimulus sequence. This preplay activity may reflect an automatic prediction mechanism for visual sequences.

## Results

**Probing preplay of stimulus sequences**. Human observers were exposed to a dot that rapidly moved across the screen (stimulation condition) from left-to-right or right-to-left for a 4 min period, while maintaining fixation (Fig. 1a). After this exposure period, occasionally only the starting point (preplay condition) or end point (no preplay condition) of the sequence was flashed, omitting the remaining dots. We reasoned that if V1 is involved in the prediction of anticipated events, cortical activity during cue-triggered preplay should resemble the activity during stimulus presentation at retinotopically defined locations while preserving the temporal order of the dot sequence. In contrast, flashing the end point is not associated with any predictions and should thus not trigger an activity wave.

Moreover, to probe the automaticity of visual preplay, we manipulated participants' attentional state. In two separate sessions, we instructed participants to either perform a covert attention task on the dot sequence (attended condition) or a demanding task at fixation (unattended condition). In the attended condition, participants had to detect rare occasions on which the last dot of the sequence was temporally delayed by 167 ms (reaction time (RT), $515 \pm 97$ ms, mean $\pm$ s.d.; error rate, $12\% \pm 10\%$). In the unattended condition, participants were presented with a sequence of rapidly changing letters at fixation and had to detect target letters (see Methods for details; RT, $419 \pm 125$ ms, mean $\pm$ s.d.; error rate, $25\% \pm 14\%$). Halfway throughout each session, the direction of the dot sequence was reversed (for example, from left-to-right to right-to-left) and participants performed the same task after 4 min of exposure to the new sequence.

**Preplay of anticipated stimulus sequences in V1**. Physically presenting the moving dot sequence triggered sequential activity at the corresponding retinotopic locations in V1 (Fig. 1b). Interestingly, briefly flashing a dot at the starting location elicited a markedly similar wave of activity. In contrast, no activity wave was observed when the end point of the sequence was flashed, ruling out the possibility that the activity wave in V1 is simply due to spatial spreading of the BOLD signal.

Next, we sought to characterize the BOLD activity trace in the preplay condition compared to the stimulus condition with respect to its amplitude and peak latency. Obviously, BOLD amplitude was higher at retinotopically defined stimulus locations receiving bottom-up input than during the preplay condition at these same locations (Fig. 1c; attended condition, two-tailed $t$-test; $t_{(28)} = 9.44$, $P = 3.39 \times 10^{-10}$; unattended condition, two-tailed $t$-test; $t_{(28)} = 4.33$, $P = 1.72 \times 10^{-4}$). Crucially, there was larger BOLD activity in these locations during preplay than during the control condition (attended condition, two-tailed $t$-test; $t_{(28)} = 16.50$, $P = 5.88 \times 10^{-16}$; unattended condition, two-tailed $t$-test; $t_{(28)} = 14.76$, $P = 9.79 \times 10^{-15}$). Further, covert attention to the dot locations led to increased BOLD amplitude (analysis of variance (ANOVA); $F_{(2,27)} = 4.70$, $P = 2.63 \times 10^{-6}$). Importantly, withdrawing attention from the stimulus to fixation reduced the BOLD amplitude in the stimulation and preplay condition to an equal amount (stimulation: 53%; preplay: 51%; stimulation versus preplay: two-tailed $t$-test; $t_{(28)} = 1.22$, $P = 0.31$), suggesting an equal amount of preplay during the attended and unattended conditions. Averaged BOLD time courses are shown in Fig. 2. Control analyses revealed that the preplay effect persists when using a variable inter-trial interval (ITI) (Supplementary Fig. 5) and when the stimulus sequence crosses through fixation (Supplementary Fig. 7). To illustrate the spatial specificity of the activity spread we performed a pRF-based reconstruction of the stimulus[10,11] based on all voxel in V1 (see Methods section). This visualization shows clearly that the temporal wave is constrained to the approximate stimulus locations (Fig. 3).

**Temporal compression of preplayed activity**. Does the activity wave during preplay propagate faster than the activity wave during stimulus presentation? Animal studies found that internally generated sequence reactivations are often temporally compressed, compared to the actual stimulus sequence[7,12]. During exposure to the moving dot sequence, we found a monotonic increase in the peak latency of the BOLD response for locations that were stimulated later, reflecting the different onset times of the dot sequence (Fig. 4a). The speed of the activity wave was quantified as the slope of the best linear fit of the BOLD peak times across the four stimulus locations. A temporal compression factor (TCF) was calculated by dividing the slope of the stimulus condition by the slope of the preplay condition (TCF > 1 indicates temporally compressed preplay). In line with earlier electrophysiological studies, we observed significant temporal compression in the attended condition (two-tailed $t$-test; $t_{(28)} = 6.45$, $P = 6.59 \times 10^{-7}$; TCF = $2.22 \pm 0.23$, mean $\pm$ s.e.m.) and in the unattended condition (two-tailed $t$-test; $t_{(28)} = 3.83$, $P = 7.01 \times 10^{-4}$; TCF = $2.29 \pm 0.34$, mean $\pm$ s.e.m.).

**Preplay amplitude in V1 correlates with hMT+ amplitude**. The prediction signal in V1 might be generated within the visual system or be the result of feedback from higher-level visual areas encoding motion such as motion-sensitive area hMT+ (refs 13,14). Indeed, hMT+ was significantly activated during both the presentation of the moving dot sequence and preplay, but not in the control condition (Fig. 4b). Interestingly, hMT+ BOLD amplitude was correlated with V1 amplitude during stimulation (attended condition, two-tailed $t$-test; $t_{(28)} = 4.41$, $P = 1.39 \times 10^{-4}$; unattended condition, two-tailed $t$-test;

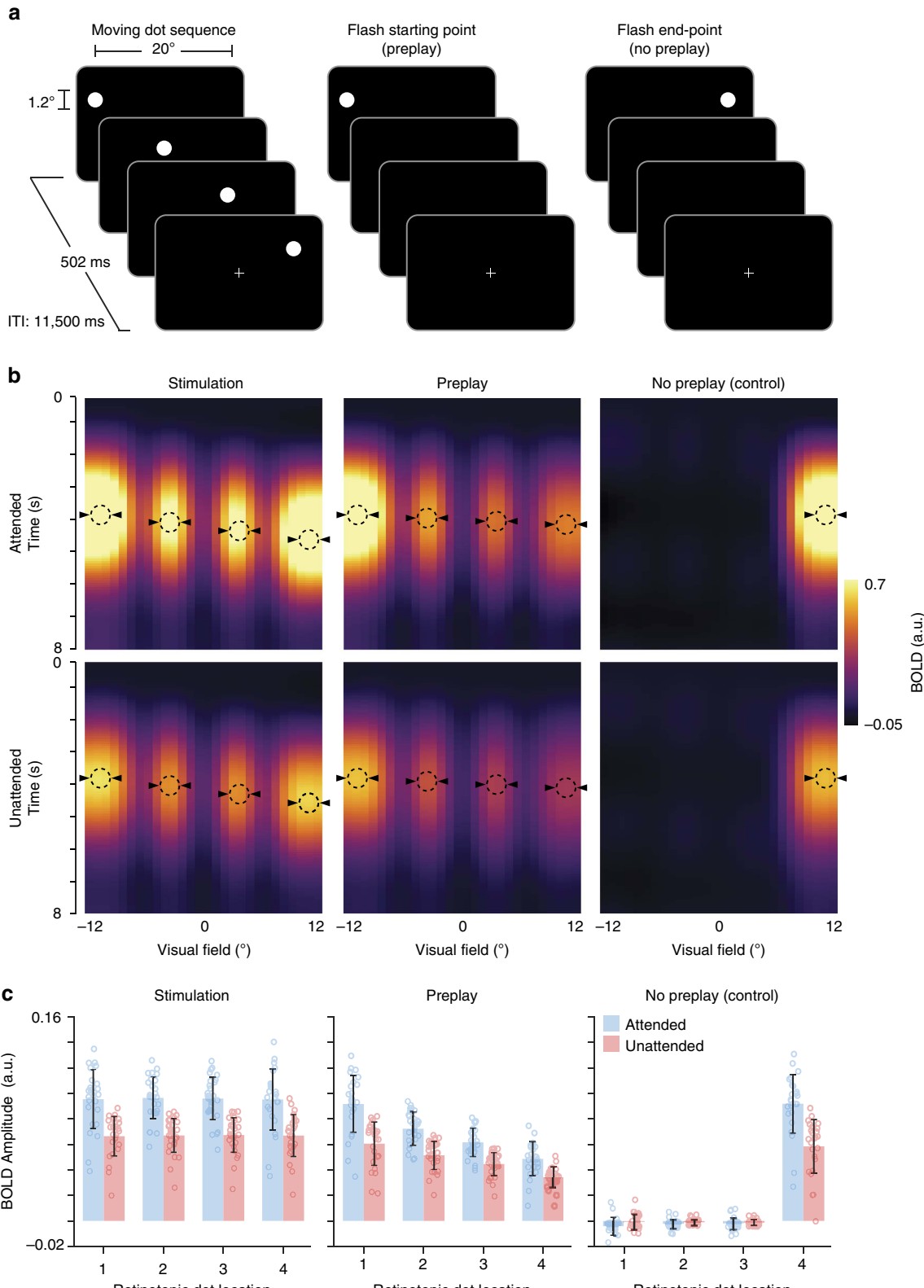

**Figure 1 | Cue-triggered activity preplay in human V1. (a)** Experimental paradigm. Participants were instructed to do a task on the dot sequence (attended condition) or at fixation (unattended condition). **(b)** Fitted BOLD responses as a function of retinotopic horizontal eccentricity during presentation of the stimulus sequence (left), preplay (middle) and no preplay (right) for the attended and unattended condition, respectively. The two different stimulus sequences, left-right and right-left, were combined by flipping the reconstruction of the right-left trials. Dashed circles depict horizontal stimulus locations. Triangles depict the BOLD peaks. **(c)** Corresponding BOLD amplitudes at the stimulus locations for attended (blue) and unattended (red) conditions. Error bars denote ± s.e.m.

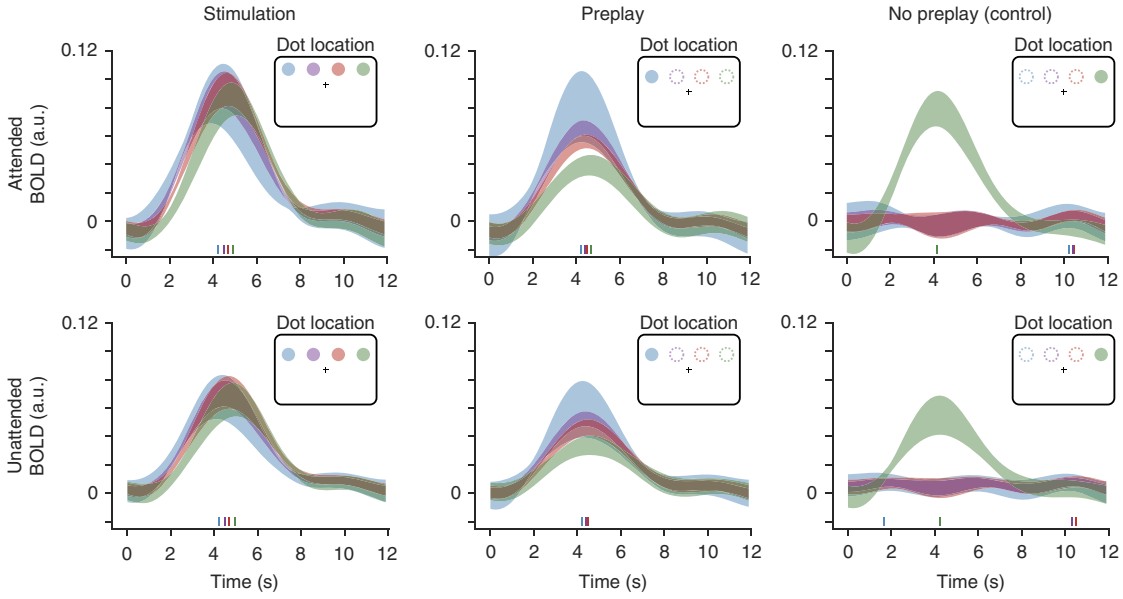

**Figure 2 | Average BOLD responses at the four stimulus locations.** BOLD response during presentation of the stimulus sequence (left), preplay (middle) and no preplay (right) for the attended and unattended conditions, respectively. The two different stimulus sequences, left-right and right-left, were combined by averaging the respective trials. Coloured lines along the time axis depict the BOLD peaks. Shaded areas denote ± s.d.

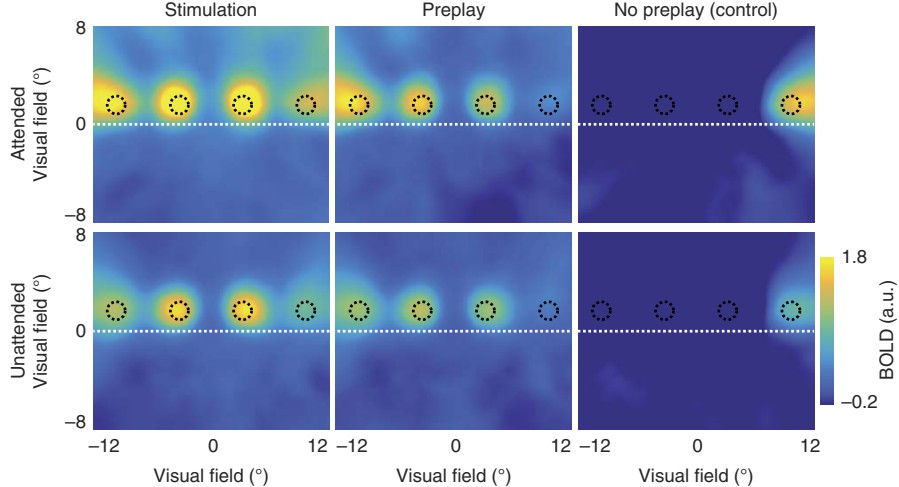

**Figure 3 | Stimulus reconstruction from BOLD activity in V1.** Reconstruction of the BOLD response evoked by stimulation, preplay and control conditions. Images were obtained by weighting all voxels' Gaussian receptive fields by the respective BOLD amplitude in each condition and then averaging these responses over all pRFs. The black circles illustrate the spatial position of the dots. The dashed white line depicts the horizontal meridian.

$t_{(28)} = 5.20$, $P = 1.59 \times 10^{-5}$) and preplay (attended, two-tailed $t$-test; $t_{(28)} = 5.15$, $P = 1.82 \times 10^{-5}$; unattended, two-tailed $t$-test; $t_{(28)} = 4.36$, $P = 1.60 \times 10^{-4}$), but not during the control condition (attended, two-tailed $t$-test; $t_{(28)} = 0.72$, $P = 0.48$; unattended, two-tailed $t$-test; $t_{(28)} = -0.01$, $P = 0.99$).

**Preplay facilitates the detection of upcoming events.** Next we examined whether the observed preplay waves might have behavioural relevance, for example, by facilitating the detection of upcoming stimulus events. To test this hypothesis, we compared RTs and BOLD peak times for the delayed sequence trials (Fig. 5a) in which participants had to respond as fast as possible when the last dot of the stimulus sequence was temporally delayed. We reasoned that if BOLD latency at the final dot position depends on the anticipation of the final dot,

faster BOLD responses may allow for faster behavioural detection of the delayed stimulus sequence. Delayed sequence trials were divided based on the median RT ($RT_{Median} = 515$ ms ± 105, mean ± s.d.) into fast and slow detection trials ($RT_{Fast} = 439$ ms ± 101, mean ± s.d.; $RT_{Slow} = 633$ ms ± 104, mean ± s.d.), separately for each participant. Results show that BOLD time courses corresponding to the delayed dot location (Fig. 5b) peaked significantly earlier for fast detection trials, compared to slow detection trials (Fig. 5c; two-tailed $t$-test; $t_{(28)} = 4.03$, $P = 4.93 \times 10^{-4}$), indicating that the preplay signal might indeed be relevant for facilitating the detection of upcoming events. A control analysis showed that differences between fast and slow detection trials were specific to the spatial location of the last dot, ruling out the possibility that the results were influenced by general factors like attentional fluctuations (Supplementary Fig. 1).

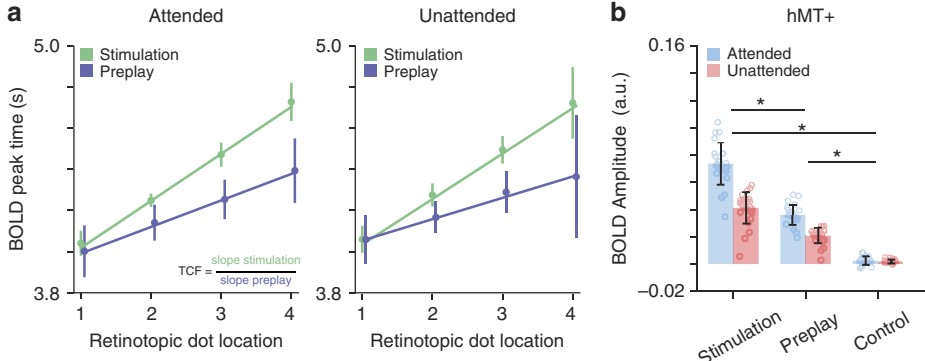

**Figure 4 | Temporal compression of activity preplay in V1.** (**a**) Peak time of the BOLD responses averaged for voxels at the four retinotopically defined stimulus locations. The fitted slope reflects the speed of the BOLD activity sequence, where a steeper slope corresponds to a temporally slower activity wave. The slopes were used to calculate the TCF for preplay compared to the stimulus sequence. (**b**) Average BOLD amplitude in hMT + for stimulation, preplay and control for the attended and unattended task conditions, respectively. Error bars denote ± s.e.m.; *$P < 0.05$.

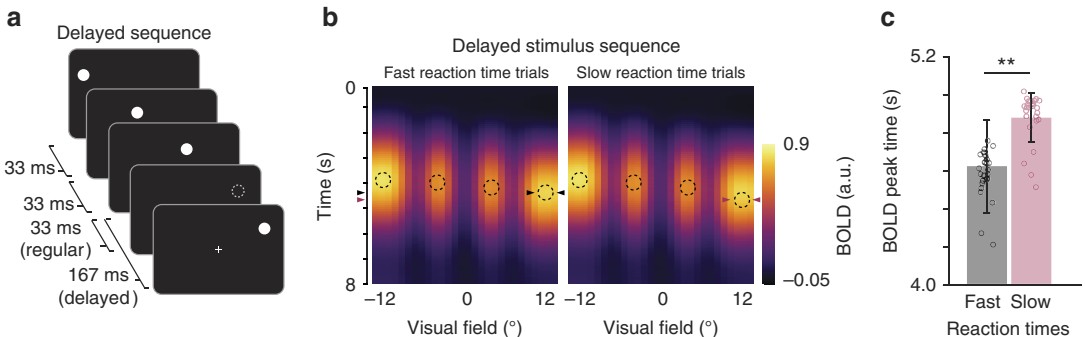

**Figure 5 | Preplay facilitates the detection of upcoming stimulus events.** (**a**) Participants had to detect delayed stimulus sequences in which the last dot occurred slightly later (167 ms) compared to the regular stimulus sequence (33 ms). (**b**) BOLD responses as a function of retinotopic horizontal eccentricity for fast (left) and slow (right) behavioural reaction times during presentation of the delayed stimulus sequence. Dashed circles depict horizontal stimulus locations. Triangles depict the BOLD peaks. (**c**) BOLD response peak significantly earlier in fast compared to slow reaction time trials. Error bars denote ± s.e.m.; **$P < 0.01$.

**Relationship between preplay and illusory motion.** Previous studies have reported elevated activity in the primary visual cortex during apparent motion[13–16], with a BOLD peak that is usually rather late compared to the stimulus onset and that appears temporally delayed compared to activity elicited by real motion (for example, see Fig. 4 in ref. 13). This is in line with the proposal that illusory motion can be understood as a post-dictive perceptual phenomenon[17,18], in which the motion percept along the apparent motion trajectory is constructed after the stimuli have been presented. It has been further reported that perception of illusory motion is abolished when the fixation point falls on the apparent motion path, potentially because of the less precise representation of stimuli in the periphery. In contrast, predictive preplay effects would be expected to remain intact when stimuli are presented near fixation, as anticipatory activity is also elicited for foveated stimuli[19,20].

To compare the anticipatory BOLD activity that we found during preplay with apparent motion, we performed a control experiment in which the first and last dots were presented alternating at 2.3 Hz for a duration of 6 s (see Methods section and Supplementary Fig. 7). This induces a percept of illusory motion between the two dot locations and has been associated with V1 activity along the path of illusory motion[13]. The apparent motion experiment contained two conditions in which the illusory motion would either pass through or above fixation.

V1 BOLD amplitude at the unstimulated apparent motion path peaked at 10 s and was significantly different from baseline activity

when the illusory motion path was above fixation (non-parametric $t$-test with 10,000 permutations; $t_{(3)} = 13.66$, $P = 8.49 \times 10^{-4}$; $BF_{10} = 36.04$), but not when it went through fixation (non-parametric $t$-test with 10,000 permutations; $t_{(3)} = 0.50$, $P = 0.65$; $BF_{01} = 2.12$). Directly comparing the BOLD activity above fixation and through fixation revealed a significant difference (non-parametric $t$-test with 10,000 permutations; $t_{(3)} = 13.56$, $P = 8.68 \times 10^{-4}$; $BF_{10} = 35.51$). Importantly, BOLD amplitude at the unstimulated dot locations during preplay through fixation was significantly higher than baseline (non-parametric $t$-test with 10,000 permutations; $t_{(3)} = 4.15$, $P = 0.03$; $BF_{10} = 3.76$).

## Discussion

Our data provide evidence for sequential activation of V1 receptive fields in response to a starting point in the same order as they appear during actual stimulation. The preplayed activity wave was temporally compressed compared to the activity wave observed during stimulation, suggesting that the preplayed activity wave reflects anticipation of future stimulus events, rather than perceptual surprise about unexpected omission of input[19,21], as the latter should occur only later in time. These results are in line with previous studies showing that memory reactivation in the hippocampus is often time-compressed[12,22,23], albeit the exact compression factor can vary[8].

Importantly, the fact that the activity wave remained present even when attention was diverted from the stimulus sequence

suggests that preplay does not strongly depend on the attentional state, but may rather reflect an automatic prediction process. This observation is consistent with the presence of preplay-like activity also in mildly anaesthetized rats[7].

Our results are based on fMRI measurements and therefore reflect haemodynamic effects. Although it has been claimed that anticipation can entrain haemodynamic responses without any corresponding neuronal activity[24], this claim has proven questionable[25]. Moreover, waves of spiking activity have been observed in awake rats in anticipation of a moving dot[7]. Therefore, it appears plausible that our effects reflect modulations of neuronal activity.

Crucially, in terms of functionality, the preplay of anticipated stimulus events that we observed in this study was associated with facilitated detection of the delayed stimulus sequence, suggestive of a link between preplay and participants' behavioural performance. An interesting question for future studies is whether preplay effectively improves the accuracy of detecting relevant stimuli along the preplayed trajectory or simply enhances the detectability of stimuli (including false positives). This could, for example, be tested by presenting only the first dot of the sequence and ask participants to perform a detection task along the preplay trajectory.

The amplitude of the preplay effect in V1 and the heightened activity in hMT + could possibly indicate a perceptual phenomenon, similar to what is observed during visual imagery[26] or illusory motion[13,14,16,27,28]. Indeed, a sizable proportion of the participants reported seeing something resembling a stripe after being exposed to only the starting location of an anticipated dot sequence. This suggests that strong anticipation of perceptual events may lead to heightened activation of relevant stimuli and stimulus locations, sometimes resulting in the misperception (sometimes called hallucination) of the corresponding visual events[29,30]. It is important however to note that imagery is an unlikely explanation for the preplay effect that was reported here. If imagery is understood as a deliberate process of recreating a visual percept[31], then it would be expected to be stronger when this process serves the participant (attended condition; task on the sequence) than when the participant is absorbed by a different task at fixation (unattended condition). Although the demanding fixation task may not fully abolish the effect of imagery, it would nevertheless be expected to be attenuated. In contrast, our data show that although drawing attention away from the dots led to a general activity reduction, preplay was of equal magnitude during both tasks (Fig. 1).

The BOLD activity spread during preplay is reminiscent of what has been found during motion-inducing illusions[15,16,28,32,33], for example, during apparent motion when stimuli are successively presented at different locations[13,14]. Does the preplay activity and the activity found during apparent motion reflect a similar neuronal mechanism? Kolers and Grunau[34] showed in a behavioural experiment that illusory motion perception strongly depends on the distance from fixation, such that distant stimuli elicit the strongest illusory motion, and the illusory motion almost fully disappears when the stimulus path crosses through fixation[34]. These behavioural findings correspond to our apparent motion experiment that revealed elevated hMT + activity and V1 activity along the apparent motion path when the stimuli were presented away from fixation, possibly reflecting the perception of illusory motion[13], but not when the motion path crossed through fixation. In contrast, the preplay effect remained present when the stimulus sequence passed through fixation.

Another difference between preplay and apparent motion may be its time course. The BOLD time course of the V1 activity along the unstimulated apparent motion path revealed a slowly rising signal, peaking after 10 s. In contrast, the time course of the

preplay activity was found to be temporally compressed, and occurred in anticipation of the sensory stimulation. These differences suggest a distinction between these phenomena, in which the apparent motion signal corresponds to a visual illusion that occurs more likely in the visual periphery with sparser receptive field coverage, and builds up slowly over time, rather than a predictive process (although it has been shown that apparent motion activity can be modulated by stimulus expectation[14]).

Perception depends on both the current sensory input and on previous experience[35]. Pattern completion, the experience-dependent ability to recreate an event based on partial information, was previously attributed to higher brain areas such as the hippocampus[36–38]. Our results extend these findings by showing that this mechanism could be partly supported by reinstating the full stimulus sequence in V1 of early visual cortex. Thus, the notion of preplay processes in the visual system blurs the boundaries between memory and perception[39,40], and underscores the integrated nature of these two cognitive faculties.

## Methods

**Participants.** Thirty-three right-handed participants were recruited from the student population at the Radboud University in Nijmegen. Sample size was decided on before the collection of data and was aimed at being able to detect experimental effects that had at least moderate effect size (Cohen's $d > 0.6$). Participants gave written informed consent in accordance with the institutional guidelines of the local ethical committee (CMO region Arnhem-Nijmegen, The Netherlands) and received monetary compensation for their participation. All participants were invited for two separate scanning sessions taking place within maximally 2 weeks' time. Three participants completed only one of the two sessions and 1 participant was excluded due to excessive head motion. Only the remaining 29 participants (19 female, mean age = 23 years) were included in all analyses. All participants had normal or corrected-to-normal visual acuity. Four additional participants (1 female, mean age = 28 years) participated in a control experiment.

**MRI acquisition.** Functional and anatomical images were acquired using a 3T Skyra MRI system (Siemens, Erlangen, Germany) equipped with a 32-channel headcoil. Each of two MRI sessions lasted between 1.5 and 2.5 h, during which we acquired (i) a T1-weighted anatomical scan (1 mm isotropic), (ii) two whole-brain functional localizer runs during the behavioural training of the stimulus sequence (2 mm isotropic; TR = 1,800 ms). These runs were used for an online analysis at the MRI scanner console to aid the slice positioning of the ultra-fast functional runs, (iii) eight ultra-fast functional runs to measure BOLD activity during the experimental paradigm (2 mm isotropic; TR = 88 ms), where four runs consisted of the left-right and right-left stimulus sequence, respectively. After four runs, participants were trained with the opposite sequence. The order of left-right and right-left sequences was randomized across sessions and participants. Functional slices were positioned based on an anatomical landmark (along the calcarine sulcus) and based on a functional localizer (stimulus versus rest) that was acquired during training. (iv) Three to four functional runs with a moving bar sequence, to estimate pRFs[9]. In one session, participants performed a task on the dot sequence (attended condition), and in the second session participants performed a task at fixation (unattended condition). The order of sessions was randomized across participants. The retinotopic mapping was only performed once, either during the first or during the second session.

BOLD activity for the localizer runs and the retinotopic mapping was measured using T2\*-weighted gradient echo planar imaging sequence (TR/echo time (TE) = 1,800/30 ms, 26 transversal slices, voxel size $2 \times 2 \times 1.7$ mm, 60° flip angle; slice gap = 20%). BOLD activity for the experimental runs was measured using a standard T2\*-weighted gradient echo planar imaging sequence (TR/TE = 88/25 ms, 2 transversal slices, voxel size $2.3 \times 2.3 \times 4.0$ mm, 60° flip angle; slice gap = 10%). Notably, the fast TR was possible because only 2 slices were acquired that were carefully positioned to cover the relevant parts of primary visual cortex (Supplementary Fig. 4). Anatomical images were acquired with a T1-weighted magnetization prepared rapid gradient echo (MP-RAGE) sequence (TR/TE = 2,300/3.03 ms, voxel size $1 \times 1 \times 1$ mm, 8° flip angle). BOLD measurements in the control experiment used a multiband sequence (acceleration factor = 3; TR/TE = 262/35.80 ms, 9 transversal slices, voxel size $2.4 \times 2.4 \times 2.4$ mm, 38° flip angle; slice gap = 20%).

**Visual stimuli.** Stimuli were rear projected on a screen located 80.5 cm from the participant's eyes at the head of the scanner table. The screen was viewed using a mirror attached to the headcoil. We presented a moving dot sequence consisting of four dots at spatial locations evenly spaced between $x = -10°$ and $+10°$ and

$y = +2°$ above fixation on a black background. Each dot had a diameter of 1.2° and was shown for 100 ms followed by a blank screen of 33 ms. The total duration of the moving dot sequence amounts to 502 ms. A fixation cross (0.6°) was shown at the centre of the screen and participants were instructed to maintain fixation throughout the experiment.

**Experimental design.** The experiment consisted of four parts. First, participants were shown 108 trials of either the dot sequence left-to-right or right-to-left. The sequences were presented in six blocks of 36 s and each trial lasted 2 s. The blocks were followed by a 36 s rest period, only displaying the fixation cross. This exposure period served to familiarize participants with the dot sequence and was further used to compute an online $t$-contrast (stimulation > fixation) at the MR console to aid the slice positioning of the following ultra-fast sequence. During this part only the full stimulus sequences were shown and no starting-point-only or end-point-only trials. During the exposure period participants performed the same task as during the main experiment, detecting letters at fixation (unattended condition) or performing a task on the dot (attended condition).

Second, there followed four runs with the ultra-fast fMRI sequence. Two slices were positioned in the sagittal axis with alignment along the calcarine sulcus, the anatomical location of V1, and adjusted, if necessary, to cover as much as possible of the online $t$-contrast map. Each run contained 24 trials, consisting of 10 stimulus sequence trials that matched the previous exposure period (that is, either left-to-right or right-to-left), 5 preplay trials with a presentation of the starting dot only, 5 control trials with a presentation of the end dot only and 4 stimulus sequence 'oddball' trials where the last dot was shown after 167 ms instead of 33 ms. The order of stimulus conditions was pseudo-randomized for a given run with the restriction that the first trial of a run was always a full sequence trial, and preplay and control trials were always preceded and followed by a full sequence trial. Each trial lasted 11.97 s (corresponding to 136 fMRI volumes), consisting of a 502 ms stimulus sequence (638 ms for the oddball trials) and 11.47 s ITI. The long ITI duration was chosen to provide time for the BOLD response to return back to baseline. Each run lasted 4.8 min and started with 8 s of fixation that was discarded from the analysis.

Third, participants were shown 108 trials of the other dot sequence, that is, left-to-right if the previous sequence was right-to-left and vice versa. Which sequence was shown first was randomly chosen and counterbalanced across participants. Fourth, after exposure with the new stimulus sequence, participants underwent four runs with the ultra-fast sequence again with the same parameters as described above. In total, the functional experiment lasted 45 min and contained 80 stimulation trials (10,880 volumes), 40 preplay trials and 40 control trials (5,440 volumes, respectively).

Each participant performed the experiment twice in separate sessions, at least 2 days, but no longer than 14 days apart. In one session (attended condition) participants had to detect rare occasions (20%), when the last dot of the sequence followed slightly later (167 ms) after the previous dot than expected (33 ms). In the other session (unattended condition), participants were presented with a sequence of rapidly changing letters at fixation and had to report whenever target letters 'X' or 'Y' (target probability = 10%) appeared in a stream of non-target letters ('A', 'T', 'N', 'U', 'V', 'Y', 'H' and 'R'). Letters were presented for 400 ms each, separated by 400 ms intervals in which only the fixation point was presented. Apart from the different tasks, the two sessions were identical. The order of sessions was counterbalanced across participants. In a post-experimental interview participants were asked whether they perceived a 'stripe' after being exposed to only the starting location of the dot sequence (attended condition: 38%; unattended condition: 28%).

A control experiment was conducted to rule out carry-over effects of the BOLD signal from one trial to the next and to test whether the preplay still persists when the stimulus sequence moves through fixation (Supplementary Fig. 7). Stimuli and timing in the control experiment were virtually identical to the main experiment. Changes were as follows: (i) instead of a fixed ITI of 11.47 s, a variable ITI ranging from 11.79 to 15.98 s was used and each block contained three null events of 10.22 s in which only a fixation cross was displayed. The null events occurred at random positions throughout the block; (ii) we presented the moving dot sequence consisting of four dots at spatial locations evenly spaced between $x = -8°$ and $+8°$ and $y = 0°$, moving through fixation; (iii) instead of the letter task, participants had to detect a dimming of the fixation cross (reduction of stimulus contrast by 30%). The dimming occurred at random locations in time, on average once per trial.

Participants were exposed to the stimulus sequence for 108 trials. Directly after that two blocks with the left-to-right sequence were shown. Each block consisted of 20 trials in which the full stimulus sequence was shown, and 10 preplay trials with the starting point only. The end-point trials were left out to keep the scanning time to a minimum.

In addition, four blocks with an apparent motion paradigm were shown in which two dots were presented alternating at 2.3 Hz. Stimulus duration was 150 ms with an inter-stimulus interval of 67 ms. The timing was based on a previous study by Muckli et al.[13]. The stimuli were shown for 6 s followed by a variable ITI ranging from 11.79 to 15.98 s. The dots were presented at $x = -8°$ and $+8°$, respectively. In two blocks, the dots were shown at $y = 6°$ and in the remaining two blocks the dots were shown at $y = 0°$, with the apparent motion path going through fixation. Each block consisted of 20 trials with the flickering apparent motion stimulus sequence, and 10 trials in which only the starting point was shown. The

latter condition was used to test whether the starting point of the apparent motion paradigm would also elicit an anticipatory BOLD activity wave.

**pRF estimation.** The data from the moving bar runs were used to estimate the pRF of each voxel in the functional volumes using MrVista (http://white.stanford.edu/software). In this analysis, a predicted BOLD signal is calculated from the known stimulus parameters and a model of the underlying neuronal population. The model of the neuronal population consisted of a two-dimensional Gaussian pRF, with parameters $x0$, $y0$ and $\sigma$, where $x0$ and $y0$ are the coordinates of the centre of the receptive field, and $\sigma$ indicates its spread (s.d.) or size. All parameters were stimulus-referred and their units were degrees of visual angle. These parameters were adjusted to obtain the best possible fit of the predicted to the actual BOLD signal. This method has been shown to produce pRF size estimates that agree well with electrophysiological receptive field measurements in monkey and human visual cortex. For details of this procedure, see refs 9,41. Once estimated, $x0$ and $y0$ were converted to eccentricity and polar-angle measures and co-registered with the functional images using linear transformation. For the following analyses, only voxels with a model fit of $R^2 > 1\%$ were considered. Group-averaged receptive field properties are shown in Supplementary Fig. 2.

**Functional localizer.** Images of the localizer blocks were preprocessed using FSL[42], including motion correction (six-parameter affine transform), temporal high-pass filtering (128 s) and spatial smoothing using a Gaussian kernel (full-width at half maximum = 5). Onsets and durations of the moving dot exposure blocks were convolved with a single-gamma haemodynamic response function (HRF) and fitted using a general linear model. For each subject a two-sided $t$-contrast was calculated comparing stimulation and baseline periods. Resulting statistical maps were thresholded at $P < 0.001$, uncorrected. Significant effects were also tested on a group-level via random-effects analysis using FSL's FLAME[43] and corrected for multiple comparisons using cluster correction with FSL's cluster command ($z = 2.58$, cluster significance threshold $P < 0.05$). In the visual cortex this resulted in separate significant clusters in hMT + and early visual cortex.

**Selection of V1 and hMT + regions.** V1 was determined using the automatic cortical parcellation provided by FreeSurfer[44] based on individual T1 images. The V1 mask was further restricted to voxels with receptive fields along the moving dot path. To this end, only voxels with a receptive field centre ranging from $y = 1.4°$ to 2.6°, and $x = -10°$ to 10° (that is, along the stimulus path) were considered (Supplementary Fig. 2b). With increasing receptive field size, voxels will respond to multiple dot locations. To prevent overlap in response profiles the V1 mask was further restricted to voxels with a pRF size ≤3.5°.

hMT + was determined for individual participants as follows. Significant voxels from the individual contrast (stimulus > baseline) were only considered when they overlapped with significant voxels in the hMT + cluster from the statistical group analysis. This way, individual hMT + activations were constrained by the group results.

**fMRI preprocessing.** Images were preprocessed using FSL (Oxford, UK) including motion correction (six-parameter affine transform, albeit limited due to low number of slices), temporal high-pass filtering (128 s) and Savitzky–Golay low-pass filter[45] for each run separately. Individual voxel time courses were then transformed into per cent signal change in reference to the mean over time. No spatial smoothing was performed, and all analyses were carried out in the native subject space.

Next, time courses were temporally averaged over trials and runs separately for each task condition (stimulus, preplay and control), stimulus sequence (left-to-right and right-to-left) and sessions (attended and unattended). In the following analyses only voxels within individual V1 and hMT + masks were considered.

**Amplitude and peak latency.** fMRI response amplitude and peak latency were computed by fitting a conventional single-gamma HRF function[46] to individual voxel time courses for each participant, task condition, stimulus sequence, session and each region of interest (ROI). This was done using the function curve_fit as implemented in SciPy 0.18 (ref. 47; based on the Trust Region Reflective algorithm for a constrained least-squares fit). The objective function was the sum of squared errors between the predicted and observed response. We allowed the baseline, amplitude and peak delay to vary as free parameters. The peak delay was constrained to a range between 3 and 11.5 s peak latency to prevent pathological fits.

Owing to cortical magnification, the receptive field density is highest around the fovea and rapidly decreases towards the periphery. As a consequence V1 voxels have disproportionally more receptive fields close to the fovea, compared to the periphery (Supplementary Fig. 2a). To give an unbiased estimate of the propagation of visual activity, we binned the receptive fields into 28, evenly spaced bins, covering the relevant eccentricities from $x = -12°$ to $+12°$. Note, the binning procedure only aids the visualization of the V1 activity wave, but had no influence on the reported BOLD amplitude and peak latency statistics. Fitted HRFs were averaged within each bin and resulted for each subject, task condition,

stimulus sequence and session in a $M = N \times T$ matrix, where $N$ is the number of bins and $T$ is the number of functional volumes (136). For the different stimulus sequences, that is, left-to-right and right-to-left, matrices $M$ were aligned by reversing the order of $N$ for the right-to-left sequences (effectively changing it into a left-to-right sequence) and then averaged. Note that due to missing pRF coverage, single bins might be empty. Importantly, while averaging these empty bins were not treated as zero, but as 'not a number' (Nan) and the averaging was restricted to real values (for example, average(5.3 + 2.1) = 7.4; average(5.3 + Nan) = 5.3; average(Nan + Nan) = Nan). Individual response matrices are shown in Supplementary Fig. 3. The relationship of BOLD amplitude and BOLD peak delay was tested using Pearson's correlation across subjects for the stimulation condition. Confirming previous reports[48,49], no significant relationship was found ($r = -0.12$; $P = 0.81$).

To what extent does the detection of the reported BOLD delay differences depend on our ultra-fast scanning sequence? To answer this question, we down-sampled the existing data and repeated the same analyses (Supplementary Fig. 6).

For the BOLD amplitude and delay analyses, only V1 voxel along the stimulus path were selected. We additionally used a pRF-based stimulus reconstruction analysis[10,11] based on all voxel in V1 with a pRF size $\leq 3.5°$, to show the spatial specificity of the activity spread. To this end, for each condition, the BOLD amplitude estimates of each voxel were first multiplied with a two-dimensional Gaussian defined by each voxels' pRF estimates ($x0$, $y0$ and $s0$) and then averaged over the voxel dimension, resulting in a stimulus reconstruction. This stimulus reconstruction was divided by a stimulus reconstruction for which the amplitude for all voxels was set to 1, effectively reducing the distortion due to the uneven distribution of receptive fields throughout the visual field. To illustrate the temporal and spatial dynamics of BOLD activity, we calculated pRF-based stimulus reconstructions ($N = 29$) for each time point (Supplementary Movie 1). These reconstructions were created for the unattended condition. The same pRF-based stimulus reconstruction analysis was used for the control experiment with apparent motion stimuli (Supplementary Fig. 7).

BOLD amplitude for hMT+ was calculated using the same fitting procedure, except that responses were averaged over all hMT+ voxels and no binning was performed. Correlations between V1 and hMT+ amplitude estimates were calculated using Pearson's product-moment correlation across subjects. To this end, the V1 amplitudes of the stimulation condition were averaged across all four stimulation locations (that is, dot locations 1–4). For the preplay and control condition V1 amplitudes were averaged across the non-stimulated dot locations (that is, dot locations 2, 3 and 4 for preplay and dot locations 1, 2 and 3 for the control condition). Event-related averages of the BOLD responses are shown in Fig. 2. Resulting correlation coefficients were transformed to Fisher $z$ values and tested for significance from zero using one sample $t$-tests (non-parametric permutation test with 10,000 permutations).

**Granger correlation of hMT+ and V1.** We used a Granger 'causality' analysis, hereby referred to as Granger correlation (GC) analysis to probe the 'directionality' of the correlation between hMT+ and V1. GC was calculated in terms of vector autoregressive models in the frequency domain 0.0078–0.25 Hz (that is, the available frequency spectrum of the BOLD signal after temporal filtering). The influence measures $F_{hMT+ \to V1}$, $F_{V1 \to hMT+}$ and $F_{hMT+ \cdot V1}$ were computed from the average time course of all voxels in hMT+ and V1 for the stimulation and preplay condition, respectively. As suggested by Roebroeck et al.[50] we assessed GC as the difference of the influence terms ($F_{hMT \to V1}$ and $F_{V1 \to hMT}$). The null hypothesis is $F_{hMT+ \to V1} - F_{V1 \to hMT+} = 0$. The order of the autoregressive model was set to $p = 1$ according to Roebroeck et al.[50].

Within the GC framework, a difference value $< 0$ can be interpreted as 'hMT+ is driving V1', and a difference value $> 0$ can be interpreted in the opposite direction, that is, that 'V1 is driving hMT+'. GC difference scores were tested against zero using a two-sided one-sample $t$-test (Supplementary Fig. 8).

**RT analysis.** On the basis of a median split, for each participant, delayed sequence trials were divided into fast versus slow RT trials. BOLD time courses of voxels corresponding to the retinotopically defined location of the last dot in the stimulus sequence were averaged and fitted with a HRF for each participant and slow versus fast RT trials separately. Estimated BOLD peak latency differences were tested across participants with a paired-sample $t$-test.

**Control for eye movements.** Participants were instructed to maintain fixation throughout the whole experiment. Eye positions were recorded with a video camera at 50 Hz sampling rate under infrared illumination (MEye Track-LR camera unit, SMI, SensoMotoric Instruments). Eyeblink artefacts were identified by differentiating the signal to detect eye pupil changes occurring too rapidly ($< 60$ ms) to represent actual dilation. Blinks and samples in which the corneal reflection was not reliably detected were removed from the signal using linear interpolation. Eyetracking data gathered in the scanner for 22 of the 29 participants during the attended session and for 21 of the 29 participants for the unattended session. We calculated the mean gaze as a function of the four stimulus locations and task conditions. Mean horizontal gaze position did not vary with stimulus position for the scanning session with 'task on stimulus' (attended condition; ANOVA,

$P = 0.58$; $N = 22$) and for the scanning session with 'task at fixation' (unattended condition; (ANOVA, $P = 0.33$; $N = 21$).

**Data availability.** All data will be made available freely on request. All code will be made available freely on request.

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

## Acknowledgements

This study was supported by the James S. McDonnell Foundation (JSMF Scholar Award for Understanding Human Cognition) and the European Union Horizon 2020 Program (ERC Starting Grant 678286, 'Contextvision') awarded to F.P.d.L.

## Author contributions

M.E., P.K and F.P.d.L. conceived and designed the experiments; M.E. collected the data and conducted the data analyses; M.E., P.K. and F.P.d.L. wrote the manuscript.

## Additional information

**Competing interests:** The authors declare no competing financial interests.

