## [Peer Review File · Nature Communications]

Reviewers' comments:

Reviewer #1 (Remarks to the Author):

In this experiment, the authors use fast acquisition fMRI to test whether activity in area V1 encodes predictions about a stimulus sequence. Specifically, after exposing participants to a stimulus sequence (a dot moving across the visual field), even presenting only the start of this sequence produces a wave of activation across the locations where a stimulus would have appeared in the sequence. In contrast, in the control condition when only the end point of the sequence is shown no such wave is measured. This is a very interesting finding and the experiment uses exciting new methodology. I don't have any major concerns but I feel there are some points that could be discussed in more detail:

1. Because the fast acquisition fMRI sequence is novel, there should really be some additional information. If there has been a previous publication presenting this pulse sequence this should probably be included. Was this an accelerated multiband sequence? If so, is it at all possible that correlations between simultaneously acquired slices contaminate the result? In general, it would be good to discuss the accuracy of this sequence.

I should note that this is not a major concern. It seems very difficult to reconcile the finding with any scanning artifact. Since transverse slices were acquired the left-to-right sequence is unlikely to be contaminated by slice time effects and - more importantly - this cannot explain why the activity wave is absent for the control condition. I just think that these questions deserve more discussion.

2. The effect observed here is somewhat reminiscent to the anticipatory hemodynamic effects claimed to have been found by Sirotin & Das (2009, Nature). I believe the claim that regular stimuli entrain hemodynamic responses without any corresponding neuronal activity has already been refuted however but this seems relevant.

3. In how far could the observed effects be related to visual imagery? Imagery has been suggested to activate early visual cortex so it is possible that imagery of the dot sequence, rather than predictive processing, would produce this result. The attention manipulation perhaps already speaks against this explanation, although I don't think a demanding fixation task necessarily abolishes the effect of imagery entirely.

4. The effects observed by the authors also seem reminiscent of the findings by Sterzer Haynes & Rees (2005, NeuroImage) and Jancke, Chavane, Naaman & Grinvald (2004, Nature). In either of these experiments, the perceptual experience of the stimulus is one of illusory motion, so it is not entirely the same but the neural response profile is similar. In fact, the Sterzer study used effective connectivity analysis to infer that feedback from MT+ to V1 causes activity in unstimulated parts of V1. Future work should perhaps test whether such feedback signals are involved in Preplay activity.

5. Related to the previous points, the response in unstimulated cortex during the preplay conditions seems surprisingly strong. Even at the most distant location the response is

around half that of the actually stimulated location. This makes me wonder further whether there could be some perceptual phenomenon like imagery or illusory motion at play.

6. The last section of the results states that Preplay is associated with detecting upcoming events. This is very interesting. However, unless I am misunderstanding something, I don't think this necessarily follows from these results. In this analysis (shown in Figure 3), the responses were related to actual stimuli. So the only part of the response pattern that could be considered Preplay in this condition is the response to the final (delayed) stimulus.

During slow response trials the peak of the response more accurately reflects the true stimulus timing. It is possible that simply due to random variation during fast reaction trials activity the response was already higher and this triggered a faster reaction. The correlation between reaction time and response latency may thus be a form of regression to the mean rather than telling us about Preplay activity.

In my view this is only a small part of the results so this issue doesn't matter much for the present study. Future studies could perhaps test whether Preplay is behaviorally relevant by presenting the preplay condition (so only flashing the first dot) and then asking participants to perform a detection task on a dot in the final position with various onset asynchronies.

7. I assume Supplementary Figure 2 shows pRF results pooled across participants? The visual field coverage seems unusually homogeneous otherwise especially around the vertical meridian which we tend to see lesser pRF density in V1 (presumably due to partial volume effects with V2 etc).

Reviewer #2 (Remarks to the Author):

This is an interesting study that uses fMRI to test for effects of preplay in the human visual cortex. Such effects have been recently reported in the rat and in the monkey, but they have not been studied in humans before. The authors use a fast fMRI sequence to test for latency effects, as well as population receptive field mapping to plot the locations of greater activity to a predictable dot movie sequence.

The results are generally consistent with the predicted effects of preplay, with greater activity observed at the retinotopic locations corresponding to the anticipated location of the stimulus. There is also a statistically significant effect of differential latency across the spatial positions of the moving dot.

However, there is an important concern that needs to be addressed. A fixed intertrial interval is used in the main experiment with a period of 12s for each trial. It is possible that the apparent time course of fMRI activity in the preplay condition arises from temporal dynamics of the hemodynamic response to a previous stimulus trial. The hemodynamic response function (HRF) consists of an initial peak followed by a post-stimulus undershoot, and this often leads to gradually increasing (or recovering) activity by 12s post-stimulus or soon thereafter. If preplay trials often follow stimulus trials, this could account for the

apparent time course of responses observed on preplay trials.

Note that this would not explain the responses found in the control condition, where the dot appears on the opposite location, but both this control stimulus and its unexpected location could elicit strong enough responses to disrupt a carryover HRF effect of the previous trial. It is documented that a restricted spatial stimulus will evoke a spatiotemporal wave in the visual cortex (e.g., work by Schira and others), so the control stimulus (and attention drawn to that location) may evoke its own spatiotemporal wave, which could counteract the effects of the HRF arising from the previous trial.

An appropriate control would be to vary the inter-trial interval in a follow-up experiment with longer delay periods, and to include some blank trials as a control. Such a control experiment would help build confidence in these presumed effects that test only one version of this experiment. It would be also useful to see the full fMRI time course for each of the four locations on preplay trials, and on control trials, for the current experiment.

Specific comments

Preplay in V1 correlates with MT activity.

What is this correlation analysis being performed? Is this a correlation of individual differences in V1 and MT measures?

Might estimates of temporal compression be affected by the fact that a physical stimulus was presented at the first location and not the other locations? If the spatiotemporal response to a single stimulus was estimated at each location, and the responses at the 2nd, 3rd and 4th locations were reduced by multiplicative scaling, would this affect estimates of the latency of the response?

Was the HRF function a single gamma function fitted to the data? Note that an undershoot may not be apparent in event-related average for the fMRI time course if the duration between successive events is too short, because there is not enough time for a full recovery to baseline.

A large number of participants were run in this study, more than is common for most studies of the visual system. How was this number decided upon, at what point in the study, and why was so much data needed for this experiment?

Reviewer #3 (Remarks to the Author):

This is an interesting manuscript that investigates responses in retinotopically mapped human visual cortex that occur in response to a single flashed dot when previously

participants have been exposed to a repeated 'prior' of continuous apparent motion starting from that flashed dot location. The authors find that retinotopic cortex on the previously exposed path are activated in an apparent temporal sequence that is argued to reflect 'preplay' of stimulus expectation.

Major concerns

1. I wonder if the authors should work harder to stress the novelty (or what they perceive as the novelty) relative to the multiple earlier reports of activation on the path of apparent motion in human retinotopic visual cortex. Aren't those reports 'preplay' of anticipated trajectories (as they are path-specific, occur in the absence of stimulation). What do the authors feel is the novelty here – the temporal specificity of the activations? Currently one of those papers is cited – I think just briefly in the Methods – and a fuller discussion would be welcome to help understand how this paper relates to the broader literature.
2. I am concerned about potential amplitude/latency trade-offs in the fitting of the HRF. Specifically for an HRF where the width is fixed but the height of the HRF is the free parameter, stronger BOLD signals could result in a function that rises faster (and thus appears to have a shorter temporal latency) because it has to reach a greater amplitude, compared to BOLD signals that are weaker. This would result in signals that are weaker (perhaps on the path of the apparent motion) appear to be temporally delayed (perhaps like a 'wave'). I understand the authors had a latency parameter for the HRF, but am not entirely convinced that this can control for amplitude/latency tradeoffs if the shape of the function remains fixed. Perhaps the authors could do a better job of convincing the reader that this is not an issue?
3. A few more methodological details would help with understanding the results. One important omission are details of the central attention-demanding task and the behavioural results. Unless I missed it, I couldn't find details of how often the letter 'X' or 'Y' actually appeared in the letter sequence (p8), how fast the 'rapidly changing letters' (p8) were actually presented and so on. Similarly I can't find any details of participants' accuracy and RT on this task. Both areas of detail are important to understand because it is strongly implied throughout the paper that this task withdrew attention and so the dot sequences were 'unattended'. Knowing task details and results is important in interpreting this claim.
4. Focusing a bit more on the claims regarding attentional state, I think these are significantly overstated. In the discussion (p5) the authors claim that because the preplay remained present when attention was diverted, 'preplay does not depend on the attentional state'. This is patently not correct – Figure 1 shows clearly that diverting attention reduces preplay amplitudes dramatically. A more nuanced statement would be better. Similarly the claim that this is an 'automatic prediction process' is not supported by the evidence unless the authors can show that the attentional diversion task fully removed attention from the stimulus. There are ROC-based approaches that can demonstrate this (e.g. see the work of Jochen Braun) but these were not employed here and so again I think a softening of this claim would be worthwhile.

5. A final example of perhaps exaggerated claims is the section entitled 'Preplay in V1 correlates with feedback signal from higher visual area hMT+' (p4). To support such a claim I would expect that some measure of a feedback signal would be presented. But this is not the case. What is correlated with V1 amplitude during preplay is the amplitude of the signal in hMT+. This of course is not a 'feedback signal' – it is simply the signal in hMT+. That signal could arise through feedforward mechanisms (and indeed probably does), and may or may not correlate with the modulatory effect of feedback projections from hMT+ to V1. This section needs dramatically revising. The authors could of course actually measure a feedback signal if they performed analyses of directional connectivity (e.g. Granger causality or DCM) on the data, as some of the studies mentioned in major point 1 have done. Could the authors undertake such analyses and directly assess the claim made in the title?

6. The abstract and introduction specifically emphasise the 'ultra-fast' aspect of the MR sequence used. However, what is actually measured is the rather slow and unchanged HRF. I was thus a bit baffled about why the strong emphasis on 'ultra-fast' was present. I would suggest either removing this emphasis or demonstrating (preferably empirically) why this was necessary in the current study to demonstrate the (temporally slow) HRF effects and could not be achieved by conventional MRI. This will be important for readers attempting replication.

7. Some more details of the data selected for use in the analyses would be helpful. There are two issues here. First, only two 2mm sagittal slices were used to collect data (p7) and it seems that because they could only 'cover as much as possible' of the retinotopic cortex then some locations (or data points) might have been missed in some participants. I would like to see details given of how much data was missed and how much was included in the analyses.

8. The second data selection issues concerns those voxels selected whose pRFs covered the stimulus sequence. If I have understood correctly, the dot trajectory went between an x axis of +10 degrees and -10 degrees and the centre of that trajectory was 2 degrees above fixation. The dot was 1.2 degrees in diameter so the path spans 1.4 to 2.6 degrees above fixation I think. Thus I don't understand p9, where the pRFs selected ranged from fixation (zero degrees) to 3.5 degrees, and from +12 to -12 degrees. Does this include pRFs that did not cover the dot trajectory? It is also difficult to reconcile these selection criteria with the scatter plots in Supplementary Figure 2, where some of the pRFs appear to be outside these criteria e.g. below fixation. Some clarification would be helpful.

9. Finally, much in the paper depends on the spatial specificity of the findings and that is partially ensured by the selection of the pRFs. Would it be helpful to present a control analysis of those pRFs NOT selected in the steps mentioned in major points 7 & 8? Those pRFs lie off the dot path and so it would be reassuring to see that there is no temporal 'wave' of activity or modulation by withdrawal of attention in these areas?

We thank the reviewers for their encouraging and insightful comments on our revised manuscript “**Time-compressed preplay of anticipated events in human primary visual cortex**” (NCOMMS-16-21012).

Below you will find a detailed point-to-point response.

Reviewer #1 (Remarks to the Author):

In this experiment, the authors use fast acquisition fMRI to test whether activity in area V1 encodes predictions about a stimulus sequence. Specifically, after exposing participants to a stimulus sequence (a dot moving across the visual field), even presenting only the start of this sequence produces a wave of activation across the locations where a stimulus would have appeared in the sequence. In contrast, in the control condition when only the end point of the sequence is shown no such wave is measured. This is a very interesting finding and the experiment uses exciting new methodology.

We thank the reviewer for his/her positive evaluation and for providing positive feedback on our manuscript.

I don't have any major concerns but I feel there are some points that could be discussed in more detail:

1. Because the **fast acquisition fMRI sequence is novel**, there should really be some additional information. If there has been a previous publication presenting this pulse sequence this should probably be included. Was this an accelerated multiband sequence? If so, is it at all possible that correlations between simultaneously acquired slices contaminate the result? In general, it would be good to discuss the accuracy of this sequence.

I should note that this is not a major concern. It seems very difficult to reconcile the finding with any scanning artifact. Since transverse slices were acquired the left-to-right sequence is unlikely to be contaminated by slice time effects and - more importantly - this cannot explain why the activity wave is absent for the control condition. I just think that these questions deserve more discussion.

We apologize for not being clear about the fMRI sequence. The sequence is in fact a standard 2D EPI sequence. We obtained the fast TR by acquiring only 2 slices that were carefully aligned to cover the relevant parts of primary visual cortex based on a real-time localizer. This is now stressed more explicitly in the method section:

BOLD activity for the experimental runs was measured using a **standard** T2*-weighted gradient-echo EPI sequence (TR/TE = 88/25 ms, 2 transversal slices, voxel size 2.3x2.3x4.0 mm, 60° flip angle; slice gap = 10 %). **Notably, the fast TR was possible because only 2 slices were acquired that were carefully positioned to cover the relevant parts of primary visual cortex (Supplementary Fig. 4).**

page 8

2. The effect observed here is somewhat reminiscent to the anticipatory hemodynamic effects claimed to have been found by Sirotin & Das (2009, Nature). I believe the claim that regular stimuli entrain hemodynamic responses without any corresponding neuronal activity has already been refuted however but this seems relevant.

We thank the reviewer for alerting us of that reference. Our study is inspired by work in rodents that has shown anticipatory waves of spiking activity in the visual system in anticipation of a visual sequence (Xu, Jiang et al. 2012). In the light of this, we think it is plausible that our effects reflect neuronal activity. Nevertheless, we agree that the Sirotin & Das (2009) paper is relevant in this context. The article is now mentioned and discussed in the revised version of our manuscript:

Our results are based on fMRI measurements and therefore reflect hemodynamic effects. Although it has been claimed that anticipation can entrain hemodynamic responses without any corresponding neuronal activity (Sirotin and Das 2009) (however see (Handwerker and Bandettini 2011), waves of spiking activity have in fact been observed in awake rats in anticipation of a moving spot (Xu, Jiang et al. 2012).

page 6

3. In how far could the **observed effects be related to visual imagery**? Imagery has been suggested to activate early visual cortex so it is possible that imagery of the dot sequence, rather than predictive processing, would produce this result. The attention manipulation perhaps already speaks against this explanation, although I don't think a demanding fixation task necessarily abolishes the effect of imagery entirely.

We thank the reviewer for this comment. We now discuss the relation to visual imagery in the manuscript.

If we understand imagery as a deliberate (wilfull) process of recreating a visual percept, then this would be expected to be stronger when this process serves the participant (dot task) than when the participant is absorbed by a task at fixation (behavioral performance: RT, 419 ± 125 mean \pm s.d; Error rate, $25\% \pm 14\%$; see a video of the task in **Supplementary Movie 1**. Although it is possible that the demanding fixation task doesn't fully abolish the effect of imagery, as the reviewer also points out, it would nevertheless be expected to be attenuated.

In contrast, our data show that preplay is of equal magnitude during both tasks. Even though *overall* BOLD activity is lower when attention is drawn away from the stimuli ($F_{(2,27)} = 4.70$, $P = 2.63 \times 10^{-6}$), the *preplay effect* is equally strong in both cases (attended condition, $t_{(28)} = 16.50$, $P = 5.88 \times 10^{-16}$; unattended condition, $t_{(28)} = 14.76$, $P = 9.79 \times 10^{-15}$; attended vs unattended, $t_{(28)} = 1.22$, $P = 0.31$). This bears resemblance to findings in rodents (Xu, Jiang et al. 2012) that observed highly similar preplay even under conditions of anaesthesia, and renders voluntary mental imagery less likely as an alternative explanation. We have added these considerations to the Results and Discussion of the paper.

Importantly, withdrawing attention from the stimulus to fixation reduces the BOLD amplitude in the stimulation and preplay condition to an equal amount (stimulation: 53%; preplay: 51%). The difference was not significant ($t_{(28)} = 1.22$, $P = 0.31$).

page 4

It is important however to note that imagery is an unlikely explanation for the preplay effect that was reported here. If imagery is understood as a deliberate process of recreating a visual percept²⁹, then it would be expected to be stronger when this process serves the participant (attended condition; task on the sequence) than when the participant is absorbed by a different task at fixation (unattended condition). Although the demanding fixation task may not fully abolish the effect of imagery, it would nevertheless be expected to be attenuated. In contrast, our data show that although drawing attention away from the dots led to a general activity reduction, preplay was of equal magnitude during both tasks (**Fig. 1**).

page 7

4. The effects observed by the authors also seem reminiscent of the findings by Sterzer Haynes & Rees (2005, NeuroImage) and Jancke, Chavane, Naaman & Grinvald (2004, Nature). In either of these experiments, the perceptual experience of the stimulus is one of illusory motion, so it is not entirely the same but the neural response profile is similar. In fact, the Sterzer study used effective connectivity analysis to infer that feedback from

MT+ to V1 causes activity in unstimulated parts of V1. Future work should perhaps test whether such feedback signals are involved in Preplay activity.

We thank the reviewer for bringing these studies to our attention. We mention and discuss these references in the revised version of our manuscript:

The amplitude of the preplay effect in V1 and the heightened activity in hMT+ could possibly indicate a perceptual phenomenon, similar to what is observed during visual imagery (Albers, Kok et al. 2013), or illusory motion (Jancke, Chavane et al. 2004, Muckli, Kohler et al. 2005, Sterzer, Haynes et al. 2006, Alink, Schwiedrzik et al. 2010, Yantis & Nakama 1998). Indeed, a sizable proportion of the participants reported seeing a “stripe” after being exposed to only the starting location of an anticipated dot sequence. This suggests that strong anticipation of perceptual events may lead to heightened activation of relevant stimuli and stimulus locations, sometimes resulting in the misperception (‘hallucination’) of the corresponding visual events (Schmack, Gomez-Carrillo de Castro et al. 2013, Pajani, Kok et al. 2015).

page 6

Prompted by the reviewer’s question, we now applied effective connectivity analysis to test for feedback signals from hMT+ to V1. The results of this analysis can be found in the response to point 5 of reviewer 3.

5. Related to the previous points, the **response in unstimulated cortex during the preplay conditions seems surprisingly strong**. Even at the most distant location the response is around half that of the actually stimulated location. This makes me wonder further whether there could be some **perceptual phenomenon like imagery or illusory motion at play**.

We agree with the reviewer’s comment that both the large amplitude of the preplay effect in V1 and the heightened activity in hMT+ may suggest a perceptual phenomenon - albeit different in nature than during imagery (*see previous point*), or illusory motion (*see comment to reviewer 3 point 1 about the relationship to apparent motion*). We thank the reviewer for bringing this up, and we have now added these considerations to the Discussion section of the manuscript (p. 6).

6. The last section of the results states that Preplay is associated with detecting upcoming events. This is very interesting. However, **unless I am misunderstanding something, I don’t think this necessarily follows from these results**. In this analysis (shown in Figure 3), the responses were related to actual stimuli. So the only part of the response pattern that could be considered Preplay in this condition is the response to the final (delayed) stimulus.

During slow response trials the peak of the response more accurately reflects the true stimulus timing. It is possible that simply due to random variation during fast reaction trials activity the response was already higher and this triggered a faster reaction. The **correlation between reaction time and response latency may thus be a form of regression to the mean rather than telling us about Preplay activity**.

In my view this is only a small part of the results so this issue doesn’t matter much for the present study. Future studies could perhaps test whether Preplay is behaviorally relevant by presenting the preplay condition (so only flashing the first dot) and then asking participants to perform a detection task on a dot in the final position with various onset asynchronies.

The reviewer correctly understood that the relationship between brain and behavior was obtained in trials in which the full sequence was shown. While it is true that this analysis relies on intra-individual variability in the speed of processing the final target, we think it nevertheless is of interest that this behavioral variability is related to neural variability in a meaningful way. Moreover, the link that we established is *not* between an initial overall larger BOLD response associated with faster RT. In fact, **Supplementary Fig. 1** shows that the BOLD response is initially (1-2.5 s) even slightly lower for the fast RT trials compared to slow RT trials, but peaks earlier (RT fast ~4.4 s; RT slow ~4.6 s).

We think that this specific effect of BOLD latency fluctuations at the final dot location *in anticipation* of the presented dot suggests the potential behavioral relevance of preplay. We have made this logic more clear in the current version of the manuscript:

In order to test this hypothesis, we compared reaction times (RT) and BOLD peak-times for the delayed sequence trials (Fig. 3a) in which participants had to respond as fast as possible when the last dot of the stimulus sequence was temporally delayed. **We reasoned that if BOLD latency at the final dot position depends on the anticipation of the final dot, faster BOLD responses may allow for faster behavioral detection of the delayed stimulus sequence.** Delayed sequence trials were divided based on the median RT ($RT_{\text{Median}} = 515 \text{ ms} \pm 105$, mean \pm SD) into fast and slow detection trails ($RT_{\text{Fast}} = 439 \text{ ms} \pm 101$, mean \pm SD; $RT_{\text{Slow}} = 633 \text{ ms} \pm 104$, mean \pm SD), separately for each participant.

page 5

We nevertheless agree with the reviewer that behavioral relevance of preplay could be tested in a more direct way, using paradigms such as the one proposed by the reviewer. We now also mention this outlook in the manuscript:

Crucially, in terms of functionality, the preplay of anticipated stimulus events that we observed in this study was associated with facilitated detection of the delayed stimulus sequence, suggestive of a link between preplay and participants' behavioral performance. An interesting question for future studies is whether preplay effectively improves the accuracy of detecting relevant stimuli along the preplayed trajectory, or simply enhances the detectability of stimuli (including false positives). This could for example be tested by presenting only the first dot of the sequence and ask participants to perform a detection task along the preplay trajectory.

page 6

7. I assume Supplementary Figure 2 shows pRF results pooled across participants? The visual field coverage seems unusually homogeneous otherwise especially around the vertical meridian which we tend to see lesser pRF density in V1 (presumably due to partial volume effects with V2 etc).

We apologize for not being clear on this. We added more information to the figure caption to state explicitly that the data represent a group average and is pooled across subjects.

Supplementary Figure 2. pRF properties and voxel selection. (a) pRF density and pRF size in V1, for the attended (top) and unattended condition (bottom) **averaged across subjects (N=29)**. (b) pRF locations in V1 (red). **Data is pooled across subjects.**

page 19

Reviewer #2 (Remarks to the Author):

This is an interesting study that uses fMRI to test for effects of preplay in the human visual cortex. Such effects have been recently reported in the rat and in the monkey, but they have not been studied in humans before. The authors use a fast fMRI sequence to test for latency effects, as well as population receptive field mapping to plot the locations of greater activity to a predictable dot movie sequence.

The results are generally consistent with the predicted effects of preplay, with greater activity observed at the retinotopic locations corresponding to the anticipated location of the stimulus. There is also a statistically significant effect of differential latency across the spatial positions of the moving dot.

We thank the reviewer for his/her positive assessment.

However, there is an important concern that needs to be addressed. A **fixed intertrial interval** is used in the main experiment with a period of 12s for each trial. It is possible that the apparent time course of fMRI activity in the **preplay condition arises from temporal dynamics of the hemodynamic response to a previous stimulus trial**. The hemodynamic response function (HRF) consists of an initial peak followed by a post-stimulus undershoot, and this often leads to gradually increasing (or recovering) activity by 12s post-stimulus or soon thereafter. If preplay trials often follow stimulus trials, this could account for the apparent time course of responses observed on preplay trials.

Note that this would not explain the responses found in the control condition, where the dot appears on the opposite location, but both this control stimulus and its unexpected location could elicit strong enough responses to disrupt a carryover HRF effect of the previous trial. It is documented that a restricted spatial stimulus will evoke a spatiotemporal wave in the visual cortex (e.g., work by Schira and others), so the control stimulus (and attention drawn to that location) may evoke its own spatiotemporal wave, which could counteract the effects of the HRF arising from the previous trial.

An appropriate control would be to **vary the inter-trial interval in a follow-up experiment with longer delay periods**, and to include some blank trials as a control. Such a control experiment would help build confidence in these presumed effects that test only one version of this experiment. It would be also useful to **see the full fMRI time course for each of the four locations on preplay trials, and on control trials, for the current experiment**.

We understand the reviewer's concern. We followed the reviewers suggestion and present newly collected data for N=4 subjects with variable ITI (12,052-15,982 ms in steps of 786 ms) and additional null-events (12,052 ms). These new data also demonstrate the presence of preplay (see new **Supplementary Fig. 5** below). Therefore we think the fixed ITI is unlikely to be the source of our observation of preplay.

Participants. Four additional participants (1 female, mean age = 28 years) participated in a control experiment.

page 8

MRI acquisition. BOLD measurements in the control experiment used a multiband sequence (acceleration factor = 3; TR/TE = 262/35.80 ms, 9 transversal slices, voxel size 2.4x2.4x2.4 mm, 38° flip angle; slice gap = 20 %).

page 8

Experimental Design. A control experiment was conducted to rule out carry-over effects of the BOLD signal from one trial to the next and to test whether the preplay still persists when the stimulus sequence moves through

fixation (**Supplementary Fig. 5**). Stimuli and timing in the control experiment were virtually identical to the main experiment. Changes were: (i) instead of a fixed ITI of 11.47 s, a variable ITI ranging from 11.79 s to 15.98 s was used and each block contained three null-events of 10.22 s in which only a fixation-cross was displayed. The null-events occurred at random positions throughout the block; (ii) We presented the moving dot sequence consisting of four dots at spatial locations evenly spaced between $x=-8^\circ$ and $x=+8^\circ$ and $y=0^\circ$, moving through fixation; (iii) Instead of the letter task, participants had to detect a dimming of the fixation cross (reduction of stimulus contrast by 30%). The dimming occurred at random locations in time, on average once per trial.

Participants were exposed to the stimulus sequence for 108 trials. Directly after that, 2 blocks with the 'left-to-right' sequence were shown. Each block consisted of 20 trials in which the full stimulus sequence was shown, and 10 preplay trials with the starting-point only. The end-point trials were left out in order to keep the scanning time to a minimum.

page 9

Supplementary Figure 5. Control experiment (N=4) with variable ITI (~12-16 s). **(a)** Experimental paradigm. Participants were instructed to detect a dimming of the fixation cross (reduction of stimulus contrast by 30%). Notably, here the path of the dot sequence crosses fixation. **(b)** Corresponding BOLD amplitudes at the stimulus locations for the stimulation (green) and preplay (red) condition. **(c)** Fitted BOLD responses as a function of retinotopic horizontal eccentricity during presentation of the stimulus sequence (*left*) and preplay (*right*). Dashed circles depict horizontal stimulus locations. Error bars denote \pm s.d.

page 23

We further added **Supplementary Fig. 7**, which shows the averaged BOLD time-courses at the four stimulus locations. The BOLD response at the non-stimulated dot locations in the preplay condition shows a clear impulse response after stimulus onset, which seems distinct from the slow dynamics one would expect from a recovering BOLD signal to baseline.

Supplementary Figure 7. Average BOLD responses at the 4 stimulus locations during presentation of the stimulus sequence (*left*), preplay (*middle*) and no preplay (*right*) for the attended and unattended condition, respectively. The two different stimulus sequences, left-right and right-left, were combined by averaging the respective trials. Colored lines along the time axis depict the BOLD peaks. Shaded areas denote \pm s.d.

page 25

Specific comments

Preplay in V1 correlates with MT activity.

What is this correlation analysis being performed? Is this a correlation of individual differences in V1 and MT measures?

We apologize for not having provided enough detail about the V1-hMT+ correlation analysis. Yes, this is indeed an analysis of individual differences. We have added this information to the Methods section of the revised version of the manuscript.

Correlations between V1 and hMT+ amplitude **estimates** were calculated using Pearson product-moment correlation across subjects. **To this end, the V1 amplitudes of the stimulation condition were averaged across all four stimulation locations (i.e., dot locations 1 to 4). For the preplay and control condition V1 amplitudes were averaged across the non-stimulated dot locations (i.e., dot locations 2, 3, 4 for preplay and dot locations 1, 2, 3 for the control condition).** Resulting correlation coefficients were transformed to Fisher-z values and tested for significance from zero using one sample *t*-tests (nonparametric permutation test with 10,000 permutations).

page 13

Was the HRF function a single gamma function fitted to the data? Note that an **undershoot may not be apparent in event-related average for the fMRI time course** if the duration between successive events is too short, because there is not enough time for a full recovery to baseline.

We indeed fitted a single gamma function. This is now stated explicitly in the methods section:

fMRI response amplitude and peak latency were computed by fitting a conventional **single** gamma HRF function²⁶ to individual voxel time-courses for each participant, task condition, stimulus sequence, session, and each ROI.

page 12

The reviewer is correct that the undershoot of the BOLD response is not clearly visible in our data. However, both the control experiment with variable ITI (new **Supplementary Fig. 5**), and the event-related average of the fMRI time-course (new **Supplementary Fig. 7**) suggest that our results are not driven by a recovery to baseline effect.

A large number of participants were run in this study, more than is common for most studies of the visual system. How was this number decided upon, at what point in the study, and why was so much data needed for this experiment?

We thank the reviewer for the opportunity to clarify the sample size selection. We made a prior decision concerning number of participants. We estimated that a sample size of ~28 participants would be desired to detect effects that had an effect size of Cohen's $d > 0.6$ with 80% power. In our case, we expected some additional dropouts since each participant was scanned twice on separate days and in view of the extended experimental session. This is how we arrived at our final sample size. Our experimental effects are highly robust as can be seen from **Supplementary Fig. 3** that shows the effects for individual participants. Nevertheless, we preferred to err on the side of caution at the start of the experiment, also given that lower sample sizes have the risk of inflating the chance of false positives (Button, Ioannidis et al. 2013). We added information about the planned sample size to the manuscript:

Thirty-three right-handed participants were recruited from the student population at the Radboud University in Nijmegen. **Sample size was decided on prior to the collection of data and was aimed at being able to detect experimental effects that had at least moderate effect size (Cohen's $d > 0.6$).**

page 8

Reviewer #3 (Remarks to the Author):

This is an interesting manuscript that investigates responses in retinotopically mapped human visual cortex that occur in response to a single flashed dot when previously participants have been exposed to a repeated 'prior' of continuous apparent motion starting from that flashed dot location. The authors find that retinotopic cortex on the previously exposed path are activated in an apparent temporal sequence that is argued to reflect 'preplay' of stimulus expectation.

We thank the reviewer for the positive comments on our manuscript.

Major concerns

1. I wonder if the authors should work harder to stress the novelty (or what they perceive as the novelty) relative to the multiple **earlier reports of activation on the path of apparent motion** in human retinotopic visual cortex. Aren't those reports 'preplay' of anticipated trajectories (as they are path-specific, occur in the absence of stimulation). What do the authors feel is the novelty here – the temporal specificity of the activations? Currently one of those papers is cited – I think just briefly in the Methods – and a **fuller discussion would be welcome to help understand how this paper relates to the broader literature.**

We fully agree that we should have discussed theoretical overlap with the apparent motion literature and apologize for the oversight. We now discuss this more thoroughly in the manuscript. Additionally, we conducted an new control analysis to compare the preplay effect with previously reported apparent motion results (**Supplementary Fig. 9**). This led us to the conclusion that there is overlap but also interesting differences between these phenomena.

The results are discussed in the Results and Discussion section of the revised version of the manuscript:

Relationship between preplay and illusory motion

Previous studies have reported elevated activity in the primary visual cortex during apparent motion¹¹⁻¹⁴, with a BOLD peak that is usually rather late compared to the stimulus onset and that appears temporally delayed compared to activity elicited by real motion (e.g., see Fig. 2 in¹¹). This is in line with the proposal that illusory motion can be understood as a 'post-dictive perceptual phenomenon'^{15,16}, in which the motion percept along the apparent motion trajectory is constructed after the stimuli have been presented. It has been further reported that perception of illusory motion is abolished when the fixation point falls on the apparent motion path, potentially because of the less precise representation of stimuli in the periphery. In contrast, predictive preplay effects would be expected to remain intact when stimuli are presented near fixation, as anticipatory activity is also elicited for foveated stimuli^{17,18}.

In order to compare the anticipatory BOLD activity that we found during preplay with apparent motion, we performed a control experiment in which the first and last dot were presented alternating at 2.3 Hz for a duration of 6 s (see Materials and Methods and **Supplementary Fig. 9**). This induces a percept of illusory motion between the two dot locations and has been associated with V1 activity along the path of illusory motion¹¹. The apparent motion experiment contained two conditions in which the illusory motion would either pass through or above fixation.

V1 BOLD amplitude at the unstimulated apparent motion path peaked at ~10 s and was significantly different from baseline activity when the illusory motion path was above fixation (nonparametric t-test with 10,000 permutations; $t(3) = 13.66$, $P = 8.49 \times 10^{-4}$; $BF_{10} = 36.04$), but not when it went through fixation (nonparametric t-test with 10,000 permutations; $t(3) = 0.50$, $P = 0.65$; $BF_{10} = 2.12$). Directly comparing the BOLD activity above fixation and through fixation revealed a significant difference (nonparametric t-test with 10,000 permutations; $t(3) = 13.56$, $P = 8.68 \times 10^{-4}$; $BF_{10} = 35.51$). Importantly, BOLD amplitude at the unstimulated dot locations during preplay through fixation was significantly higher than baseline (nonparametric t-test with 10,000 permutations; $t(3) = 4.15$, $P = 0.03$; $BF_{10} = 3.76$).

page 5

Experimental Design. Additionally, 4 blocks with an apparent motion paradigm were shown in which two dots were presented alternating at 2.3 Hz. Stimulus duration was 150 ms with an inter-stimulus interval of 67 ms. The timing was based on a previous study by Muckli et. al. (2005). The stimuli were shown for 6 s followed by a variable ITI ranging from 11.79 s to 15.98 s. The dots were presented at $x=-8^\circ$ and $x=+8^\circ$ respectively. In 2 blocks, the dots were shown at $y=6^\circ$ and in the remaining 2 blocks the dots were shown at $y=0^\circ$, with the apparent motion path

going through fixation. Each block consisted of 20 trials with the flickering apparent motion stimulus sequence, and 10 trials in which only the starting-point was shown. The latter condition was used to test whether the starting point of the apparent motion paradigm would also elicit an anticipatory BOLD activity wave.

page 10

Supplementary Figure 9. Control analysis (N=4) contrasting our results with an apparent motion paradigm. **(a)** Stimulus sequence and preplay through fixation (*top*). Group average of the pRF-based reconstruction (see *Materials and Methods*) from V1 BOLD activity (*bottom*). **(b)** Apparent motion paradigm above fixation and through fixation (*top*). Group average of the pRF-based stimulus reconstruction from V1 BOLD activity. **(c)** BOLD time-courses along the apparent motion path (*white box*) for V1 (*top*) and hMT+ (*bottom*) for the apparent motion paradigm. The group averaged V1 BOLD response shows a peak at around 10 s after stimulus onset. Dashed circle highlights the position of the respective stimulus positions. Error bars denote \pm s.e.m.

page 27

Discussion. The BOLD activity spread during preplay is reminiscent of what has been found during motion-inducing illusions^{13,14,26,30,31}, e.g. during apparent motion when stimuli are successively presented at different locations^{11,12}. Does the preplay activity and the activity found during apparent motion reflect a similar neuronal mechanism? Kolers & Grunau (1977) showed in a behavioral experiment that illusory motion perception strongly depends on the distance from fixation, such that distant stimuli elicit the strongest illusory motion and the illusory motion almost fully disappears when the stimulus path crosses through fixation³². These behavioral findings correspond to our apparent motion experiment that revealed elevated hMT+ activity and V1 activity along the apparent motion path when the stimuli were presented away from fixation, possibly reflecting the perception of illusory motion¹¹, but not when the motion path crossed through fixation. In contrast, the preplay effect remained present when the stimulus sequence passed through fixation.

Another difference between preplay and apparent motion may be its time course. The BOLD time-course of the V1 activity along the unstimulated apparent motion path revealed a slowly rising signal, peaking after ~10 s. In contrast, the time-course of the preplay activity was found to be temporally compressed, and occurred in anticipation of the sensory stimulation. These differences suggest a distinction between these phenomena, in which the apparent motion signal corresponds to a visual illusion that occurs more likely in the visual periphery with sparser receptive field coverage, and builds up slowly over time, rather than a predictive process (although it has been shown that apparent motion activity can be modulated by stimulus expectation¹²).

page 7

2. I am concerned about potential **amplitude/latency trade-offs in the fitting of the HRF**. Specifically for an HRF where the width is fixed but the height of the HRF is the free parameter, stronger BOLD signals could result in a function that rises faster (and thus appears to have a shorter temporal latency) because it has to reach a greater amplitude, compared to BOLD signals that are weaker. This would result in **signals that are weaker** (perhaps on the path of the apparent motion) **appear to be temporally delayed** (perhaps like a 'wave'). I understand the authors had a latency parameter for the HRF, but am not entirely convinced that this can control for amplitude/latency tradeoffs if the shape of the function remains fixed. Perhaps the authors could do a better job of convincing the reader that this is not an issue?

The reviewer is correct that BOLD onset *latency* and BOLD amplitude have been reported to be weakly correlated (Thompson, Engel et al. 2014), with higher BOLD amplitude being associated with earlier onset times. Importantly however, no such relationship was found for BOLD *peak time* and amplitude (Lindquist and Wager 2007, Casanova, Ryali et al. 2008, Thompson, Engel et al. 2014). To empirically verify whether this relationship was also absent in our data, we correlated BOLD amplitude with the BOLD peak time across participants during the stimulation condition. As expected, there was no reliable relationship between these metrics ($r = -0.12$, $P = 0.81$). We have added this

The relationship of BOLD amplitude and BOLD peak delay was tested using Pearson correlation across subjects for the stimulation condition. Confirming previous reports (Lindquist and Wager 2007, Thompson, Engel et al. 2014), no significant relationship was found ($r = -0.12$; $P = 0.81$).

page 13

Finally, if weaker signals would be *temporally delayed*, as the reviewer suggested might be the case, this could not explain our preplay results, as we find the weaker preplay signal to be *temporally faster* than during visual stimulation.

3. A few more **methodological details** would help with understanding the results. One important omission are **details of the central attention-demanding task** and the **behavioural results**. Unless I missed it, I couldn't find details of how often the letter 'X' or 'Y' actually appeared in the letter sequence (p8), how fast the 'rapidly changing letters' (p8) were actually presented and so on. Similarly I can't find any details of participants' accuracy and RT on this task. Both areas of detail are important to understand because it is strongly implied throughout the paper that this task withdrew attention and so the dot sequences were 'unattended'. Knowing task details and results is important in interpreting this claim.

We apologize for omitting these important methodological details. We have now added the information to the revised version of the manuscript.

In the other session (unattended condition), participants were presented with a sequence of rapidly changing letters at fixation and had to report whenever target letters 'X' or 'Y' (**target probability = 10%**) appeared in a **stream of non-target letters ('A', 'T', 'N', 'U', 'V', 'Y', 'H', 'R')**. **Letters were presented for 400 ms each, separated by 400 ms intervals in which only the fixation point was presented.**

page 10

In the attended condition, participants had to detect rare occasions on which the last dot of the sequence was temporally delayed by 167 ms [**Reaction time (RT), 515 ± 97 ms mean ± s.d; Error rate, 12% ± 10%**]. In the unattended condition, participants were presented with a sequence of rapidly changing letters at fixation and had to detect target letters (see Methods for details; **RT, 419 ± 125 ms mean ± s.d; Error rate, 25% ± 14%**).

page 3

To illustrate the task, we now made a video depicting the task along with the fMRI results (**Supplementary Movie 1**).

4. Focusing a bit more on the **claims regarding attentional state, I think these are significantly overstated**. In the discussion (p5) the authors claim that because the preplay remained present when attention was diverted, 'preplay does not depend on the attentional state'. **This is patently not correct** – Figure 1 shows clearly that diverting attention reduces preplay amplitudes dramatically. A more nuanced statement would be better. Similarly the claim that this is an '**automatic prediction process**' **is not supported** by the evidence unless the authors can show that the attentional diversion task fully removed attention from the stimulus. There are ROC-based approaches that can demonstrate this (e.g. see the work of Jochen Braun) but these were not employed here and so again I think a softening of this claim would be worthwhile.

We thank the reviewer for bringing this up. The reviewer correctly points out that diverting attention to fixation dramatically reduces the amount of activity at the four dot locations (by on average 53%). However, this attentional effect is equally present for the stimulated dot location (reduction of 55%) and for the unstimulated dot locations (reduction of 51%), and therefore appears an additive effect on top of the preplay effect. To formally test this, we compared the effect of the attentional modulation between stimulation and preplay, and found that they are not significantly different ($t = 1.22$, $P = 0.31$). We believe that this supports our claim that the magnitude of preplay is not affected by the attentional state, even though attention itself strongly modulates the BOLD response. We now made this distinction between BOLD amplitude and the preplay effect more clearly in the result section of the revised version of the manuscript.

Importantly, withdrawing attention from the stimulus to fixation reduced the BOLD amplitude in the stimulation and preplay condition to an equal amount (stimulation: 53%; preplay: 51%; stimulation vs. preplay: $t_{(28)} = 1.22$, $P = 0.31$), suggesting an equal amount of preplay during the attended and unattended condition.

page 4

We also followed the reviewers' suggestion to use more nuanced wording in statements relating to the automatic prediction process. As an example:

Importantly, the fact that the activity wave remained present even when attention was diverted from the stimulus sequence suggests that preplay does not strongly depend on the attentional state, but may rather reflect an automatic prediction process.

page 6

5. A final example of perhaps exaggerated claims is the section entitled 'Preplay in V1 correlates with feedback signal from higher visual area hMT+' (p4). To support such a claim I would expect that some measure of a feedback signal would be presented. But this is not the case. What is correlated with V1 amplitude during preplay is the amplitude of the signal in hMT+. **This of course is not a 'feedback signal' – it is simply the signal in hMT+. That signal could arise through feedforward mechanisms (and indeed probably does)**, and may or may not correlate with the modulatory effect of feedback projections from hMT+ to V1. **This section needs dramatically revising**. The authors could of course actually measure a feedback signal if they performed analyses of directional connectivity (e.g. Granger causality or DCM) on the data, as some of the studies mentioned

in major point 1 have done. Could the authors undertake such analyses and directly assess the claim made in the title?

We agree with the reviewer that these claims were overstated and not supported by the data. We have changed the wording so that it is clear now that we measure and interpret amplitude-to-amplitude correlations and removed all references related to directionality. For example:

Preplay **amplitude** in V1 correlated with **signal amplitude** from higher-level visual area hMT+
 The prediction signal in V1 **might be generated within the visual system, or be** the result of feedback from higher-level visual areas encoding motion such as motion-sensitive area hMT+ (Muckli et al., 2005; Alink et al., 2010).

page 5

We also followed the reviewers' suggestion to perform a Granger "causality" (GC) analysis, based on the implementation described in (Roebroeck, Formisano et al. 2005) and Goebel et al. (2003).

Revision Figure 1. Probing the directionality of the hMT+, V1 correlation using Granger 'causality' (GC) analysis. GC was calculated in terms of vector autoregressive models in the frequency domain 0.0078-0.25 Hz (i.e. the available frequency spectrum of the BOLD signal after temporal filtering). The influence measures $F_{hMT \rightarrow V1}$, $F_{V1 \rightarrow hMT}$, and $F_{hMT \cdot V1}$, were computed from the average time-course of all voxels in hMT and V1, for the stimulation and preplay condition, respectively. As suggested by Roebroeck (2005) we assessed the granger directionality as the difference of the influence terms ($F_{hMT \rightarrow V1}$, $F_{V1 \rightarrow hMT}$). The null hypothesis is $F_{hMT \rightarrow V1} - F_{V1 \rightarrow hMT} = 0$. The order of the autoregressive model was set to $p=1$ according to Roebroeck (2005). A directionality value < 0 means that hMT is 'driving' V1 (green shaded area), a directionality value > 0 suggests the opposite, i.e., that V1 is 'driving' hMT (red shaded area). Granger directionality was tested against zero using a two-sided one-sample t -test. The results show that hMT is 'driving' V1 in the preplay condition (attended: $t(23) = -3.50$; $P = 0.002$; unattended: $t(25) = -2.68$; $P = 0.013$), but not in the stimulation condition (attended: $t(23) = 1.97$; $P = 0.059$; unattended: $t(25) = -0.22$; $P = 0.829$). The difference of the granger directionality between stimulation and preplay condition was tested using a two-sample t -test (attended: $t(23) = 3.38$; $P = 0.002$; unattended: $t(25) = -2.26$; $P = 0.033$).

These results support our original interpretation that hMT+ is ‘driving’ V1 during the preplay condition, but not in the stimulus condition. However, given the controversy around GC in the fMRI context (Friston 2009, Friston 2011, Roebroeck 2011) and unresolved methodological issues associated with directionality in fMRI (David et al. 2008; Smith 2011, (Logothetis 2008)), as for example the influence of inter-regional variability in the shape of the HRF (Handwerker 2012; Seth 2013, Seth 2015), we are not convinced that the results in **Revision Fig. 1** are free of such confounds. Furthermore, initial exploration of free parameters of the GC analysis, like the model order and the frequency range, suggest that the results vary considerably based on these parameters; although no exhaustive parameter search was performed. For these reasons, we would prefer not to include the GC analysis results in the revised version of the manuscript. If the reviewer would strongly urge us to include them, we would nevertheless be willing to oblige.

6. The abstract and introduction specifically emphasise the ‘ultra-fast’ aspect of the MR sequence used. However, what is actually measured is the rather slow and unchanged HRF. I was thus a bit baffled about why the strong emphasis on ‘ultra-fast’ was present. I would suggest **either removing this emphasis or demonstrating (preferably empirically) why this was necessary in the current study to demonstrate the (temporally slow) HRF effects** and could not be achieved by conventional MRI. This will be important for readers attempting replication.

The reviewer is correct that BOLD is a rather slow signal. However, it is also a temporally precise signal, which means that small latency differences in neuronal activity carry over to produce small latency differences in hemodynamic activity. Indeed, earlier fMRI studies have reported that small timing differences can be detected reliably when using fast sampling rates (Menon, Luknowsky et al. 1998, Lee, Blake et al. 2005). It is however much more difficult to reliably detect such latency differences when using a slower sampling rate. Here, we run a simulation to illustrate the dependency of BOLD latency detectability and our ultra-fast fMRI approach. The results of our simulation are shown in **Revision Fig. 2** (for similar results see also Fig. 3 in (Roebroeck, Formisano et al. 2005)).

Revision Figure 2. Detectability of BOLD latency differences depends on sampling rate (TR) and noise level. We generated two canonical HRFs with a delay latency of 100 ms. The HRFs were sampled with different sampling rates, ranging from 88 ms (the TR in our manuscript) to 3 s, and fitted with a single gamma HRF, identical to the

approach employed in our manuscript. The latency error was calculated as the absolute difference of the true latency (100 ms) and the estimated latency. Ideally the estimated latency error should be close to zero. Since the noise level of the BOLD signal was not known a priori, we further varied the noise level from $\sigma = 0$ (no noise) to $\sigma = 0.6$ for an auto-regressive process with an order of 1 (simulating the auto-correlated noise in fMRI data). Each simulation was repeated 1,000 times and latency errors were averaged across these iterations. The results show a strong dependency on the noise level and sampling rate. Within our analysis framework, for a TR of ~ 2.2 s the noise levels needs to be unrealistically low (virtually zero) in order to detect a BOLD latency of 100 ms. In comparison, for our TR of 0.088, the noise level can be ~ 8 -times higher and our analysis will still be able to detect the 100 ms latency difference. Dashed isolines depict the threshold at which the BOLD latency detection becomes significant with $\alpha = 0.05$ and 0.01 respectively.

We also followed the reviewer's suggestion to empirically demonstrate the necessity of our ultra-fast fMRI sequence for the detection of the reported BOLD latency effects (**Supplementary Fig. 8**). To this end, we down-sampled the existing fMRI data and demonstrate that the known stimulus latency between two dots can only be accurately estimated with a sampling rate < 200 ms.

To what extent does the detection of the reported BOLD delay differences depend on our ultra-fast scanning sequence? In order to answer this question, we down-sampled the existing data and repeated the same analyses (**Supplementary Fig. 8**).

page 13

Supplementary Figure 8. Detection of BOLD latency differences depends on fast fMRI sampling rate. (a) Existing fMRI data (TR = 88 ms) were down-sampled to illustrate the dependency of detecting relatively short stimulus latencies on sampling rate. (a) The BOLD latency error (closer to zero is better) was calculated as the absolute difference between the known stimulus latency (133 ms) and the estimated BOLD latency for two dot stimuli during the presentation of the stimulus sequence. (b) BOLD latency error increases with sampling rate. Statistical significance is based on one-sample *t*-tests comparing the estimated BOLD latency between the two stimuli across participants against zero. In the tested range, only sampling rates < 200 ms allowed for an accurate estimation of the known stimulus latency. Error bars denote \pm s.d.; * = $P < 0.05$.

page 26

7. Some more details of the data selected for use in the analyses would be helpful. There are two issues here. First, only two 2mm sagittal slices were used to collect data (p7) and it seems that because they could only 'cover

as much as possible' of the retinotopic cortex then some locations (or data points) might have been missed in some participants. I would like to see details given of how much data was missed and how much was included in the analyses.

We thank the reviewer for this comment. In the revised version of this manuscript, we added an additional information and a figure to illustrate the coverage:

Supplementary Figure 4. Results of online localizer to aid slice positioning of the ultra-fast scanning sequence. GLM z-statistic contrasting stimulus sequence compared to baseline, superimposed on a standard brain template; averaged for the attended and unattended condition (left). Dashed white lines highlight the approximate slices thickness and coverage of the voxels active during the online localizer (right). Across subjects, the 2 slices covered 64% (28% SD) of the activated voxels in V1 and 31% (12% SD) in hMT; respectively averaged over attended and unattended condition.

page 22

8. The second data selection issues concerns those **voxels selected whose pRFs covered the stimulus sequence**. If I have understood correctly, the dot trajectory went between an x axis of +10 degrees and -10 degrees and the centre of that trajectory was 2 degrees above fixation. The dot was 1.2 degrees in diameter so the path spans 1.4 to 2.6 degrees above fixation I think. Thus I don't understand p9, where the pRFs selected ranged from fixation (zero degrees) to 3.5 degrees, and from +12 to -12 degrees. Does this include pRFs that did not cover the dot trajectory? It is also **difficult to reconcile these selection criteria with the scatter plots in Supplementary Figure 2**, where some of the pRFs appear to be outside these criteria e.g. below fixation. Some clarification would be helpful.

We thank the reviewer for bringing this to our attention and we apologize for not being clear about the voxel selection. The reviewer is correct that the voxels with a receptive field center from 1.4 to 2.6 degrees above fixation (corresponding to the stimulus path) were selected for the analysis. However, taking into account the size of the receptive field (i.e., not only the center), the average trajectory area extends beyond 1.4 to 2.6 degrees (e.g. a voxel with $x_0, y_0 = 0, s_0 = 2.5$ covers activity from below $y = 1.4$ degrees). Our previous specification ($y = 0.5$ to 3.5 degrees) took that into account. We agree that this was indeed confusing and was removed it from the revised version of the manuscript.

The V1 mask was further restricted to voxels with receptive fields along the moving dot path. To this end, only voxels with a receptive field **center** ranging from $y = 1.4^\circ$ to $y = 2.6^\circ$, and $x = -10^\circ$ to $x = 10^\circ$ (i.e., **along the stimulus path**) were considered.

page 11

The scatter plot in **Supplementary Fig. 2** visualized the pRF-center in red and the extent of the pRF with a grey circle around the center. We believe that this might have given the impression that pRFs outside the stimulus path were selected. In the revised version of this manuscript **Supplementary Fig. 2** was changed to illustrate the voxel selection more clearly.

Supplementary Figure 2. (b) pRF locations in V1 (*red*). Data is pooled across subjects. The green box depicts the voxels along the stimulus path ($y=1.4^\circ$ to $y=2.6^\circ$, and $x=-10^\circ$ to $x=10^\circ$) that were selected for the analysis.

page 19

9. Finally, much in the paper depends on the spatial specificity of the findings and that is partially ensured by the selection of the pRFs. Would it be helpful to present a **control analysis of those pRFs NOT selected in the steps mentioned in major points 7 & 8**? Those pRFs lie off the dot path and so it would be reassuring to see that there is no temporal 'wave' of activity or modulation by withdrawal of attention in these areas?

We thank the reviewer for this thoughtful comment. In order to illustrate the spatial specificity of the findings we performed a prf-based reconstruction of the stimulus based on *all voxel* in the visual field. This visualization shows clearly that the temporal wave is constrained to the approximate stimulus locations.

For the BOLD amplitude and delay analyses, only V1 voxel along the stimulus path were selected. We additionally employed a pRF-based stimulus reconstruction analysis (Kok & de Lange 2014) based on *all voxel* in V1 with a pRF-size $\leq 3.5^\circ$, in order to show the spatial specificity of the activity spread. To this end, for each condition, the BOLD amplitude estimates of each voxel were first multiplied with a 2D Gaussian defined by each voxels' pRF estimates (x_0 , y_0 , s_0) and then averaged over the voxel dimension, resulting in a stimulus reconstruction. This stimulus reconstruction was divided by a stimulus reconstruction for which the amplitude for all voxels was set to one, effectively canceling out the distortion due to the uneven distribution of receptive fields throughout the visual

field. The same pRF-based stimulus reconstruction analysis was used for the control experiment with apparent motion stimuli (**Supplementary Fig. 9**).

page 13

Supplementary Figure 9. Stimulus reconstruction from BOLD activity in V1. Reconstruction of the BOLD response evoked by stimulation, preplay and control conditions. Images were obtained by weighting all voxels' Gaussian receptive fields by the respective BOLD amplitude in each condition and then averaging these responses over all pRFs. The black circles illustrate the spatial position of the dots. The dashed white line depicts the horizontal meridian.

page 24

Other changes

The y-axis of **Fig. 1c** and **Fig. 2b** were incorrectly labeled as ranging from -0.2 to 0.16. The correct range is -0.02 to 0.16. The typo was corrected in the revised version of the manuscript. This typo had no influence on the visualization or statistics of the data.

References

- Bullier, J., J. M. Hupe, A. C. James and P. Girard (2001). "The role of feedback connections in shaping the responses of visual cortical neurons." *Prog Brain Res* **134**: 193-204.
- Button, K. S., J. P. Ioannidis, C. Mokrysz, B. A. Nosek, J. Flint, E. S. Robinson and M. R. Munafò (2013). "Power failure: why small sample size undermines the reliability of neuroscience." *Nat Rev Neurosci* **14**(5): 365-376.
- Casanova, R., S. Ryali, J. Serences, L. Yang, R. Kraft, P. J. Laurienti and J. A. Maldjian (2008). "The impact of temporal regularization on estimates of the BOLD hemodynamic response function: a comparative analysis." *Neuroimage* **40**(4): 1606-1618.

18

Handwerker, D. A. and P. A. Bandettini (2011). "Hemodynamic signals not predicted? Not so: a comment on Sirotin and Das (2009)." Neuroimage **55**(4): 1409-1412.

Jancke, D., F. Chavane, S. Naaman and A. Grinvald (2004). "Imaging cortical correlates of illusion in early visual cortex." Nature **428**(6981): 423-426.

Lee, S. H., R. Blake and D. J. Heeger (2005). "Traveling waves of activity in primary visual cortex during binocular rivalry." Nature Neuroscience **8**(1): 22-23.

Lindquist, M. A. and T. D. Wager (2007). "Validity and power in hemodynamic response modeling: a comparison study and a new approach." Hum Brain Mapp **28**(8): 764-784.

Logothetis, N. K. (2008). "What we can do and what we cannot do with fMRI." Nature **453**(7197): 869-878.

Menon, R. S., D. C. Luknowsky and J. S. Gati (1998). "Mental chronometry using latency-resolved functional MRI." Proc Natl Acad Sci U S A **95**(18): 10902-10907.

Muckli, L., N. Kriegeskorte, H. Lanfermann, F. E. Zanella, W. Singer and R. Goebel (2002). "Apparent motion: event-related functional magnetic resonance imaging of perceptual switches and States." J Neurosci **22**(9): RC219.

Pajani, A., P. Kok, S. Kouider and F. P. de Lange (2015). "Spontaneous Activity Patterns in Primary Visual Cortex Predispose to Visual Hallucinations." J Neurosci **35**(37): 12947-12953.

Roebroeck, A., E. Formisano and R. Goebel (2005). "Mapping directed influence over the brain using Granger causality and fMRI." Neuroimage **25**(1): 230-242.

Schmack, K., A. Gomez-Carrillo de Castro, M. Rothkirch, M. Sekutowicz, H. Rossler, J. D. Haynes, A. Heinz, P.

Petrovic and P. Sterzer (2013). "Delusions and the role of beliefs in perceptual inference." J Neurosci **33**(34): 13701-13712.

Sirotin, Y. B. and A. Das (2009). "Anticipatory haemodynamic signals in sensory cortex not predicted by local neuronal activity." Nature **457**(7228): 475-479.

Thompson, S. K., S. A. Engel and C. A. Olman (2014). "Larger neural responses produce BOLD signals that begin earlier in time." Front Neurosci **8**: 159.

Xu, S., W. Jiang, M. M. Poo and Y. Dan (2012). "Activity recall in a visual cortical ensemble." Nature Neuroscience **15**(3): 449-455, S441-442.

REVIEWERS' COMMENTS:

Reviewer #1 (Remarks to the Author):

The authors have comprehensibly addressed all my previous comments. The additional control experiments also further enhance their findings. The only remaining issue I noticed is that the preplay condition in Suppl. Fig. 5b is referred to as red in the caption but appears grey to me. (Perhaps this is a PDF conversion glitch?)

I forgot to sign my previous review,
Sam Schwarzkopf, UCL

Reviewer #2 (Remarks to the Author):

The revised manuscript addresses my concerns. Supplementary Figure 7 is particularly helpful and convincing. This is an interesting study and the effect of preplay is surprisingly large.

Reviewer #3 (Remarks to the Author):

This is an excellent revision. I am grateful to the authors for considering and responding to my suggestions. The manuscript is substantially improved and I have no remaining concerns. With respect to the issue of whether to include the Granger Causality analysis, I am happy to leave this to the authors' discretion.

We thank all the reviewers for their positive evaluation of our revised manuscript “Time-compressed preplay of anticipated events in human primary visual cortex” (NCOMMS-16-21012A).

Below you will find a point-to-point response.

Reviewer #1 (Remarks to the Author):

The authors have comprehensibly addressed all my previous comments. The additional control experiments also further enhance their findings. The only remaining issue I noticed is that the preplay condition in Suppl. Fig. 5b is referred to as red in the caption but appears grey to me. (Perhaps this is a PDF conversion glitch?)

I forgot to sign my previous review,
Sam Schwarzkopf, UCL

We thank the reviewer for noticing this typo. The figure caption was corrected to grey.

Reviewer #2 (Remarks to the Author):

The revised manuscript addresses my concerns. Supplementary Figure 7 is particularly helpful and convincing. This is an interesting study and the effect of preplay is surprisingly large.

We have moved Supplementary Figure 7 to the main text as Figure 2.

Reviewer #3 (Remarks to the Author):

This is an excellent revision. I am grateful to the authors for considering and responding to my suggestions. The manuscript is substantially improved and I have no remaining concerns. With respect to the issue of whether to include the Granger Causality analysis, I am happy to leave this to the authors' discretion.

We thank the reviewer for her/his comments.